# Methods of Hidden Periodicity Discovering for Gearbox Fault Detection

**DOI:** 10.3390/s21186138

**Published:** 2021-09-13

**Authors:** Ihor Javorskyj, Ivan Matsko, Roman Yuzefovych, Oleh Lychak, Roman Lys

**Affiliations:** 1Department of Methods and Facilities for Acquisition and Processing Diagnostic Signals, Karpenko Physico-Mechanical Institute of National Academy of Sciences of Ukraine, 5 Naukova Str., 79060 Lviv, Ukraine; matskoivan@gmail.com (I.M.); roman.yuzefovych@gmail.com (R.Y.); oleh.lychak@ipm.lviv.ua (O.L.); 2Institute of Telecommunications and Computer Science, Bydgoszcz University of Science and Technology, 85796 Bydgoszcz, Poland; 3Department of Electronics and Computer Technologies, Ivan Franko National University of Lviv, 1 Universytetska Str., 79000 Lviv, Ukraine; roman.lys@lnu.edu.ua; 4Department of Applied Mathematics, Lviv Polytechnic National University, 12 Bandera Str., 79013 Lviv, Ukraine

**Keywords:** vibration signal, wind turbine gearbox, periodically correlated random processes, estimation techniques, amplitude spectrum, fault detection indicators

## Abstract

It is shown that the models of gear pair vibration, proposed in literature, are particular cases of the bi-periodically correlated random processes (BPCRPs), which describe its stochastic recurrence with two periods. The possibility of vibration and analysis within the framework of BPCRP approximation, in the form of periodically correlated random processes (PCRPs), is grounded and the implementation of vibration processing procedures using PCRP techniques, which are worked out by the authors, is given. Searching for hidden periodicities of the first and the second orders was considered as the main issue of this approach. The estimation of the non-stationary period (basic frequency) allowed us to carry out a detailed analysis of the deterministic part, the covariance structure of the stochastic part, and to form, using their parameters, the sensitive indicators for fault detection. The results of the processing of the wind turbine gearbox vibration signals are presented. The amplitude spectra of the deterministic oscillations and the time changes of the stochastic part power for different fault stages are analyzed. The most efficient indicators, which are formed using the amplitude spectra for practical applications, are proposed. The presented approach was compared with known in literature cyclostationary analysis and envelope techniques, and its advantages are shown.

## 1. Introduction

The vibration signals of rotating machinery are characterized by their rhythmic variety, whose key features are cyclic recurrence and stochasticity. Non-linear effects occur in machinery behavior as faults appear and, consequently, the interaction of these features is observed in vibration signal properties. This interaction is quantitatively characterized by the parameters describing the periodical or almost periodical time variation of the moment functions of the first and the second order of the cyclostationary random processes [1,2,3,4]. These processes are also called periodically or almost periodically correlated random processes [5,6,7,8,9]. Therefore, it is advisable to choose these parameters for the construction of the indicators for fault detection [10,11,12,13,14,15,16,17]. Gear pair vibration is excited by two main factors, namely, the periodic variation of teeth stiffness, due to the meshing phase, and manufacturing errors. The manufacturing errors include constant and variable step errors of the teeth. The periodic variation of the mesh stiffness causes the appearance of the harmonic components of the mesh frequency $f_{m}=rf_{1}=nf_{2}$ and its multiples. Here $f_{1}$ and $f_{2}$ are the rotation frequencies of the wheels and r and n are natural numbers. The variable error of the meshing step and the misalignment of the axes and shafts are manifested by the occurrence of harmonics with rotation frequencies $kf_{1}$ and $lf_{2}$, and also combination frequencies $pf_{m}+kf_{1}$, $pf_{m}+lf_{2}$, where p, k, and l are integer numbers. In addition, the direct spectra of vibration signal can include the components that belong to some frequency band around the resonance frequency of the gear pair in the case of a vibro-impact regime occurring.

The techniques proposed in [12,18] for gear pair vibration analysis were based on the transmission error model that was considered in [19]:(1)$$x\left(\theta \right)=x_{e}\left(\theta \right)\left[W+x_{m}\left(\theta \right)+x_{1}\left(\theta \right)+x_{2}\left(\theta \right)\right]$$
where W is a constant load and θ=θ(t) is an angular position of the gear. The terms $x_{m}\left(\theta \right)$ and $x_{e}\left(\theta \right)$ describe the contact properties of the gears, while terms $x_{1}\left(\theta \right)$ and $x_{2}\left(\theta \right)$ are caused by manufacturing error. It is supposed that in each term $x_{i}\left(\theta \right)$, $i=\bar{1,\;2}$ is periodic with a rotation period $P_{i}=1/f_{i}$ of the corresponding gear. There are three periodic terms in (1), namely $x_{e}\left(\theta \right)\left[W+x_{m}\left(\theta \right)\right]$, $x_{e}\left(\theta \right)x_{1}\left(\theta \right)$, and $x_{e}\left(\theta \right)x_{2}\left(\theta \right)$, which are periodic functions with the period $P_{m}=1/f_{m}$, $P_{1}$, and $P_{2}$. The model in the form of the cyclostationary process proposed in [12,18] was obtained by introducing a random variable modeling for the fluctuations of the angular position of the gears. The mean function of this random process includes the harmonic components of frequencies $f_{m}$, $f_{1}$, and $f_{2}$. The covariance function includes three kinds of harmonics, namely, the harmonics with frequencies that are a linear combination of the rotation frequencies $kf_{1}+lf_{2}$, the harmonics of the mesh frequency $nf_{m}$, and the harmonics with frequencies that are a linear combination of the mesh frequency and the rotation frequencies, i.e., $nf_{m}+kf_{i}$. The first and the second order non-stationarities were substantiated by the processing of vibration signals, measured on the gear systems [12,18], and the quantities that describe the structure of the cyclostationarity, estimated by means of synchronous averaging, were proposed for use in fault detection.

In [20,21,22], after applying the synchronous averaging with the period $P_{1}$ or $P_{2}$, the vibration signal was expressed as:(2)$$g\left(t\right)=\sum_{l=1}^{M}A_{l}\left[1+a_{l}\left(t\right)\right]\cos\left(2\pi f_{l}t+b_{l}\left(t\right)+\varphi_{l}\right)$$
where M is the number of gear mesh harmonics, and $A_{l}$ and $\varphi_{l}$ are the amplitude and the phase of the $l^{th}$ harmonic, respectively. The modulation effects are described by the functions $1+a_{l}\left(t\right)$ and $b_{l}\left(t\right)$, which are periodic with the considered rotation period. These functions are closely approximated to the signal deterministic component corresponding to one revolution of the selected gear. Proceeding from (2), some techniques were proposed in the literature [20,21,22] for the improvement of the analysis effectiveness. One of these consisted of the elimination from (2) of the harmonics with the tooth meshing frequency and its multiples. The residual signal often shows an evidence of faults more clearly than (2). The effective technique for the detection of local faults, such as a fractured tooth, is a filtration of (2) around the $l^{th}$ gear mesh harmonic and the analysis of its amplitude and phase modulation function [20,21,22].

In [19,23] the gear vibration signal was modeled as:(3)$$x\left(\theta \right)=x_{1}\left(\theta \right)+x_{2}\left(\theta \right)+x_{1,2}\left(\theta \right)+x_{c}\left(\theta \right)+n\left(\theta \right)$$
where $x_{1}\left(\theta \right)$ and $x_{2}\left(\theta \right)$ describe the deterministic periodic oscillations generated by the rotation of the output and input wheels, respectively, $x_{1,2}\left(\theta \right)$ is a periodic component with common period $P_{12}=r_{1}P_{1}+r_{2}P_{2}$, $x_{c}\left(\theta \right)$ is the second order cyclostationary process with period $P_{12}$, and n(θ) is a fluctuation component. The deterministic part of the signal (3) can be extracted by means of synchronous averaging with the common period $P_{12}$ of the shafts as far as it is possible [19]. The results of the experimental data processing conducted by the authors supported their assumption that the power of the random component was negligible in comparison with the power of the deterministic component.

The models for gearbox vibration proposed in the literature can be considered as particular cases of its representation in the form of BPCRPs [9,24,25,26]. The mean and the covariance functions of these processes are bi-periodic time functions. The mean function describes the modulation interaction of the deterministic oscillations, and the covariance function describes the interaction of the stochastic components. The Fourier series of the mean and covariance functions consist of the harmonics of the wheels rotation frequencies and their multiples and combinations. The harmonics of the mesh frequencies are the individual harmonics of the BPCRP representation. The concrete harmonics compositions of the deterministic and the stochastic oscillations depend on the degree of the development of a fault and its location.

The estimation of the whole complex of BPCRP characteristics of the first and second order on the basis of experimental data may be laborious and time-consuming, so it is advisable, when possible, to mitigate the issues associated with fault detection using the parameters of the BPCRP approaches. In the present paper, we show that, in the case of the appearance of a fault developed on only one of the wheels, the PCRP approach can be used. The efficiency of the PCRP technique for early fault detection and the analysis of fault growth are illustrated by the results of processing experimental data acquired at different faulty stages of a wind turbine gearbox. This work is based on the original results obtained by authors concerning the discovery and analysis of the hidden periodicities described by PCRP and their generalizations [24,25,26,27,28,29,30,31,32,33,34,35,36,37,38,39,40,41].

The main sources of novelty in this paper are as follows:The properties of the gearbox vibration model, in the form of BPCRPs, are analyzed, and the possibility of using PCRP approximation for fault diagnosis is explored;Methods of searching for hidden periodicities of the first and the second order are used for gearbox vibration analysis;The efficiency of the least square (LS) method for the estimation of the period for the vibration deterministic component and the time variation of the stochastic part power is shown;The main steps of an algorithm for gearbox vibration analysis using PCRP for fault diagnosis are given;The amplitude spectra of deterministic oscillations and the time variation power of the stochastic component are given as the characteristic features for fault stages;The most sensitive indicator for fault detection is based on the results of natural data processing.

The paper consists of the introduction, three sections divided into subsections, and the conclusions. The mathematical model of the vibration of coupled gears in the form of BPCRP and its harmonic series representation are considered in Section 2.1. The particular cases of the BPCRP model, which follow from its harmonic representation, are given in Section 2.2. The opportunity to analyze the BPCRP properties within the PCRP approximation framework is analyzed in Section 3. The peculiarities of the PCRP analysis of the vibration are considered and the main stages of this approach are briefly characterized: namely, the analysis in the stationary approximation, the detection and the analysis of the hidden periodicities of the first and the second orders, and the estimation of the covariance and spectral functions. The indicators for the detection of a fault, formed on the basis of the mean function and the variance amplitude spectrum, are determined. The results of the natural data processing are given in Section 4. For the different stages of the fault evolution, the search for the hidden periodicities using the statistics of the first and the second order was carried out. The main features of the amplitude and the power spectra of the vibration are analyzed, and the numeric values of the indicator formed on their basis for the growing fault are given. A comparison of the sensitivities of the indicators of the first and the second order was conducted. The dependences of the Fourier coefficients of the covariance function, the so-called covariance components, on time lag are considered. In the discussion section, the different techniques of the vibration analysis are compared.

## 2. BPCRP as a Model of Gear Pair Vibration Signal

### 2.1. Covariance and Spectral Functions

The efficiency of cyclostationarity signal processing techniques in machinery condition monitoring can be explained generally by their ability to reveal modulations caused by the occurrence of faults. The modulation effects in the vibration model in the form of the periodically correlated random processes (PCRP), which describe the stochastic recurrence with one period, are characterized by the jointly stationary random processes $\xi_{k}\left(t\right)$ in their harmonic representation [8,9,27]:$$\xi \left(t\right)=\sum_{k\in Z}\xi_{k}\left(t\right)e^{ik\frac{2\pi }{P_{1}}t}$$
where Z is a set of integer numbers and $P_{1}$ is a non-stationarity period (the rotation period for one of the wheels). Generalizing this representation, we may conclude that the modulation of two stochastic rhythms, induced by the rotation of two wheels, can be modeled as:(4)$$\xi \left(t\right)=\sum_{k\in Z}\xi_{k}^{\left(P_{2}\right)}\left(t\right)e^{ik\frac{2\pi }{P_{1}}t}$$
where the harmonic of frequency $\frac{2\pi }{P_{1}}$ and its multiples are modulated by PCRP with period $P_{2}$:$$\xi_{k}^{\left(P_{2}\right)}\left(t\right)=\sum_{l\in Z}\xi_{kl}\left(t\right)e^{il\frac{2\pi }{P_{2}}t}$$

Then, for the random process (4), we have:(5)$$\xi \left(t\right)=\sum_{k,l\in Z}\xi_{kl}\left(t\right)e^{i\Lambda_{kl}t}$$
where $\xi_{kl}\left(t\right)$ are jointly stationary random processes and $\Lambda_{kl}=k\left(2\pi /P_{1}\right)+l\left(2\pi /P_{2}\right)$. As can be seen, process (5) is a sum of the amplitude and phase modulated harmonics in which frequencies $\Lambda_{kl}$ are the linear combination of the two main frequencies $\Lambda_{10}=k2\pi /P_{1}$ and $\Lambda_{01}=l2\pi /P_{2}$. The mathematical expectations of the modulating processes $m_{kl}=E\xi_{kl}\left(t\right)$ are the Fourier coefficients of the mean function:(6)$$m\left(t\right)=E\xi \left(t\right)=\sum_{k,l\in Z}m_{kl}e^{i\Lambda_{kl}t}$$

For the covariance function $R\left(t,\tau \right)=E\overset{\circ }{\xi }\left(t\right)\overset{\circ }{\xi }\left(t+\tau \right)$, $\overset{\circ }{\xi }\left(t\right)=\xi \left(t\right)-m\left(t\right)$, we have:(7)$$R\left(t,\tau \right)=\sum_{k,l\in Z}R_{kl}(\tau )e^{i\Lambda_{kl}t}$$
where,
(8)$$R_{kl}\left(\tau \right)=\sum_{p,q\in Z}r_{p-k,q-l,p,q}e^{i\Lambda_{pq}\tau }$$
and $r_{pqkl}\left(\tau \right)=E\bar{\overset{\circ }{\xi_{pq}}}\left(t\right)\overset{\circ }{\xi_{kl}}\left(t+\tau \right)$, $\overset{\circ }{\xi_{pq}}\left(t\right)=\xi_{pq}\left(t\right)-m_{pq}$ are the cross-covariance functions of the modulating processes, and the “¯” signifies complex conjugation. Thus, the Fourier coefficients of the covariance function (7) are defined by the cross-covariance functions of the modulating processes in which the numbers are shifted by k and l, respectively. It follows from (8) that cross-correlations of modulating processes $\overset{\circ \;}{\xi_{kl}}\left(t\right)$ of the different numbers lead to the bi-periodical non-stationarity of the second order. The consequence of these correlations is the further correlation of the corresponding spectral components, which are quantitatively characterized by the Fourier transformation of expression (8):(9)$$f_{kl}\left(\omega \right)=\frac{1}{2\pi }\int_{-\infty }^{\infty }R_{kl}\left(\tau \right)e^{-i\omega \tau }d\tau$$

It follows from (8) that:$$f_{kl}\left(\omega \right)=\sum_{p,q\in Z}f_{p-k,q-l,p,q}\left(\omega -\Lambda_{pq}\right)$$
where
$$f_{pqkl}\left(\omega \right)=\frac{1}{2\pi }\int_{-\infty }^{\infty }r_{pqkl}\left(\tau \right)e^{-i\omega \tau }d\tau$$
are the cross-spectral densities of the modulating processes $\xi_{pq}\left(t\right)$. The Equations (8) and (9) are called the covariance and spectral components [9,24,25], respectively.

The zero^th^ covariance component $R_{00}\left(\tau \right)$ is determined by auto-covariance functions $r_{pq}\left(\tau \right)=E\bar{\overset{\circ }{\xi_{pq}}}\left(t\right)\overset{\circ }{\xi_{pq}}\left(t+\tau \right)$:$$R_{00}\left(\tau \right)=\sum_{p,q\in Z}r_{pq}\left(\tau \right)e^{-i\Lambda_{pq}\tau }$$

This is an averaged in time covariance function of random process (5), i.e., the covariance function of its the stationary approximation.

The zero^th^ spectral component
(10)$$f_{00}\left(\omega \right)=\sum_{p,q\in Z}f_{pq}\left(\omega -\Lambda_{pq}\right)$$
is a power spectral density of the stationary approximation for (5). It defines the spectral decomposition of the averaged in time instantaneous power R(0,t) for the oscillations. The random processes, the mean, and the covariance functions, which are bi-periodical functions and can be represented by series (6) and (7), are called BPCRP.

The Fourier coefficients of the covariance function and spectral density are the total characteristics of the amplitude and the phase modulation of the BPCRP carrier harmonics. The zero^th^ spectral component, as can be seen from (10), is a sum of the power of the spectral densities of the modulating processes $\xi_{pq}\left(t\right)$ shifted by $\Lambda_{pq}$. The spectral component $f_{kl}\left(\omega \right)$ (9) is a sum of the shifted cross-spectral densities for modulating processes, the numbers of which differ by numbers k and l, respectively. Proceeding from the above-mentioned assertions, we may conclude that the zero^th^ spectral function $f_{00}\left(\omega \right)$ describes the spectral composition of the oscillations and the non-zero^th^ functions $f_{kl}\left(\omega \right)$ describe the correlations of the harmonics of this composition in which the frequencies are shifted by $\Lambda_{kl}=k\left(2\pi /P_{1}\right)+l\left(2\pi /P_{2}\right)$. These correlations do not equal zero only if the modulating processes of the corresponding numbers are mutually correlated.

### 2.2. The Simplest Particular Cases

Proceeding from (5), we can quite easily obtain some particular cases of the bi-rhythmic hidden periodicity:

(a) If $\xi_{kl}\left(t\right)=c_{kl}+\eta_{kl}\left(t\right)$, where $\eta_{kl}\left(t\right)$ are uncorrelated stationary random processes and $c_{kl}$ are some complex numbers, we have an additive model:$$\xi \left(t\right)=\sum_{k,l\in Z}c_{kl}e^{i\Lambda_{kl}t}+\sum_{k,l\in Z}\eta_{kl}e^{i\Lambda_{kl}t}=s\left(t\right)+\eta \left(t\right)$$
where s(t) is a bi-periodical function and η(t) is a stationary random process with the covariance function:$$R\left(\tau \right)=\sum_{k,l\in Z}r_{kl}^{\left(\eta \right)}\left(\tau \right)e^{i\Lambda_{kl}\tau }$$
where $r_{kl}^{\left(\eta \right)}\left(\tau \right)=E\bar{\overset{\circ }{\eta_{k}}}\left(t\right)\overset{\circ }{\eta_{l}}\left(t+\tau \right)$. If $c_{kl}=0$, ∀k≠0, and ∀l≠0, then s(t) is a sum of two periodic functions:$$s\left(t\right)=\sum_{k\in Z}c_{k0}e^{ik\frac{2\pi }{P_{1}}t}+\sum_{l\in Z}c_{0l}e^{il\frac{2\pi }{P_{2}}t}$$

(b) If we put $\xi_{kl}\left(t\right)=c_{kl}\eta \left(t\right)$, where η(t) is a stationary random process, then we obtain a multiplicative model:(11)$$\xi \left(t\right)=\eta \left(t\right)\sum_{k,l\in Z}c_{kl}e^{i\Lambda_{kl}t}=\eta \left(t\right)s\left(t\right)$$

The mean function of (11), $m\left(t\right)=m_{\eta }s\left(t\right)$, $m_{\eta }=E\eta \left(t\right)$, and the covariance function:$$R\left(t,\tau \right)=R_{\eta }\left(\tau \right)s\left(t\right)s\left(t+\tau \right)$$
where $R_{\eta }\left(\tau \right)=E\overset{\circ }{\eta }\left(t\right)\overset{\circ }{\eta }\left(t+\tau \right)$, varies bi-periodically in time.

(c) In the case of $\xi_{kl}\left(t\right)=c_{kl}+\eta_{k0}\left(t\right)+\eta_{0l}\left(t\right)$, where $\eta_{k0}\left(t\right)$ and $\eta_{0l}\left(t\right)$ are jointly stationary random processes, the additive model is in the form of a sum of the bi-periodical function and two PCRPs with periods $P_{1}$ and $P_{2}$:$$\xi \left(t\right)=s\left(t\right)+\sum_{k\in Z}\xi_{k0}\left(t\right)e^{ik\frac{2\pi }{P_{1}}t}+\sum_{l\in Z}\xi_{0l}\left(t\right)e^{il\frac{2\pi }{P_{2}}t}=s\left(t\right)+\xi_{1}\left(t\right)+\xi_{2}\left(t\right)$$

(d) For $\xi_{kl}\left(t\right)=c_{k0}\xi_{0l}\left(t\right)$, we obtain a model of the amplitude modulation of the deterministic carrier by PCRP;

(e) We obtain the model in the form of a product of two PCRPs with different periods $P_{1}$ and $P_{2}$ in the case of $\xi_{kl}\left(t\right)=\xi_{k0}\left(t\right)\xi_{0l}\left(t\right)$:$$\xi \left(t\right)=\sum_{k\in Z}\xi_{k0}\left(t\right)e^{ik\frac{2\pi }{P_{1}}t}\sum_{l\in Z}\xi_{0l}\left(t\right)e^{il\frac{2\pi }{P_{2}}t}=\xi_{1}\left(t\right)\xi_{2}\left(t\right)$$

(f) If the stationary random processes $\xi_{l}\left(t\right)$ are mutually uncorrelated, then we have the product of stationary random process:$$\eta \left(t\right)=\sum_{l\in Z}\xi_{l}\left(t\right)e^{il\frac{2\pi }{P_{2}}t}$$
and PCRP:$$\xi \left(t\right)=\eta \left(t\right)\xi_{1}\left(t\right)$$

(g) The last considered model is the quadrature model or Rice representation. We obtain it in the case when $\xi_{kl}\left(t\right)=0$ and ∀k,l≠−1, 1. Assuming that:$$\xi_{1,1}\left(t\right)=\frac{1}{2}\left[\xi_{1,1}^{c}\left(t\right)-i\xi_{1,1}^{s}\left(t\right)\right],\;\xi_{1,-1}\left(t\right)=\frac{1}{2}\left[\xi_{1,-1}^{c}\left(t\right)-i\xi_{1,1}^{s}\left(t\right)\right],\;\mathrm{and}\;\xi_{-1,-1}\left(t\right)=\bar{\xi_{1,1}}\left(t\right),\;\xi_{-1,1}\left(t\right)=\bar{\xi_{1,-1}}\left(t\right)$$

Then,
(12)$$\xi \left(t\right)=\xi_{1,1}^{c}\left(t\right)\cos\left(2\pi \left(f_{1}+f_{2}\right)t\right)+\xi_{1,1}^{s}\left(t\right)\sin\left(2\pi \left(f_{1}+f_{2}\right)t\right)+\xi_{1,-1}^{c}\left(t\right)\cos\left(2\pi \left(f_{1}-f_{2}\right)t\right)+\xi_{1,-1}^{s}\left(t\right)\sin\left(2\pi \left(f_{1}-f_{2}\right)t\right)$$

Introducing the random process
$$\xi_{c}\left(t\right)=\left[\xi_{1,1}^{c}\left(t\right)+\xi_{1,-1}^{c}\left(t\right)\right]\cos\left(2\pi f_{1}t\right)+\left[\xi_{1,1}^{s}\left(t\right)-\xi_{1,-1}^{s}\left(t\right)\right]\sin\left(2\pi f_{1}t\right)$$
$$\xi_{s}\left(t\right)=\left[\xi_{1,1}^{s}\left(t\right)+\xi_{1,-1}^{s}\left(t\right)\right]\cos\left(2\pi f_{2}t\right)\;\;+\left[\xi_{1,1}^{c}\left(t\right)-\xi_{1,-1}^{c}\left(t\right)\right]\sin\left(2\pi f_{2}t\right)$$
we can re-write equation (12) in the form:$$\xi \left(t\right)=\xi_{c}\left(t\right)\cos\left(2\pi f_{1}t\right)+\xi_{s}\left(t\right)\sin\left(2\pi f_{1}t\right)$$

The quadrature components of the Rice representation are jointly PCRP.

Now, we compare the BPCRP representation (5) and the particular cases given above with the models of gear pair vibration considered in the introduction. It was mentioned previously that the deterministic part of the vibration consists of the harmonics of the mesh frequency and their multiples, linear combinations of the rotation frequencies, and the linear combination of each rotation frequency and mesh frequency. Since $f_{m}=rf_{1}=nf_{2}$, then all these frequencies belong to the set $\left\{kf_{1}+lf_{2}:\;k,l\in Z\right\}$. The Fourier coefficients of the BPCRP mean function, which describe the deterministic part of the vibration, define the complex amplitude of the corresponding harmonics. The coefficients $m_{l0}$ and $m_{0n}$ are the amplitudes of the harmonics for the additive components $s_{1}\left(t\right)$ and $s_{2}\left(t\right)$ with periods $P_{1}$ and $P_{2}$:$$s_{1}\left(t\right)=\sum_{k\in Z}m_{k0}e^{ik\frac{2\pi }{P_{1}}},\;s_{2}\left(t\right)=\sum_{l\in Z}m_{0l}e^{il\frac{2\pi }{P_{2}}}$$

Herewith, $m_{lk,0}$ is the sum of the amplitudes for the $r^{th}$ mesh harmonic and the amplitude of the ${\left(rk\right)}^{th}$ input wheel rotation harmonic and the amplitude for the ${\left(nl\right)}^{th}$ output wheel rotation harmonic. Note that the sum $s\left(t\right)=s_{1}\left(t\right)+s_{2}\left(t\right)$ has the common period $P=rP_{1}=nP_{2}$, and can be represented in the Fourier series form. However, the harmonics of this series have only a formal mathematical interpretation. The frequencies for some of them, as it follows from $kf=k/rP_{1}=k/nP_{2}$, may coincide with the rotation frequencies only in the case when the ratios k/r and k/n are the natural numbers.

The modulation interactions of the deterministic components are defined by amplitudes $m_{lk}$.

Proceeding from the above consideration, we should represent the mean and the covariance function of the gear pair vibration signal in the form of the general series (6) and (7). The covariance components $R_{k0}\left(\tau \right)$ and $R_{0l}\left(\tau \right)$ are the Fourier coefficients of the additive covariance terms. The covariance components $R_{lk}\left(\tau \right)$ characterize the modulation covariance interactions. The series (6) and (7) can be specified if the experimental data measured on the concrete gears system are analyzed by means of adequate processing techniques developed on the basis of the general model (5). It is evident that the analysis results can be employed for the verification of the particular cases described above.

The BPCRP mean and covariance function can be calculated on the basis of experimental data using the coherent (synchronous averaging) and component methods and also the LS method. Using the synchronous averaging, we can separate and analyze only the deterministic or the stochastic components of one of the two periods. The coherent methods, in many cases, cannot be used for the processing of the raw data, since the non-stationary periods, as a rule, are not an integer number and the interpolation of the data is required. Therefore, it is advisable to use the component and the LS techniques for data processing as they do not require an interpolation procedure.

The component and the LS estimators are formed as trigonometric polynomials, the coefficients of which are calculated on the basis of Fourier transforms, represented in the form of the integral sums. The period values in these transforms can be arbitrary, and only the sampling interval must satisfy some inequalities to avoid aliasing errors.

## 3. Gear Fault Detection as PCRP Estimation Issue

The calculation of all the parameters that characterize the structure of the additive and multiplicative components of the BPCRP mean and covariance function may be relatively laborious, especially in the case when the value of one period is appreciably in excess of the other. A similar situation also occurs when the fault detection problem can be solved on the basis of a smaller number of parameters. Let us consider the case when we can reduce this issue to PCRP estimation.

It was mentioned above that the BPCRP spectral composition is defined by the zero^th^ spectral component (10). Assume that $\Lambda_{10}\geq s\Lambda_{01}$, where s is some natural number. Making to pass the BPCRP signal through the linear filter with the transfer function,
$$H\left(\omega \right)=\left\{ \begin{array}{l} 1,\;\omega \in \left[\Lambda_{0L_{1}^{\left(2\right)}},\;1.9\omega_{m}\right],\\ 0,\;\omega \notin \left[\Lambda_{0L_{1}^{\left(2\right)}},\;1.9\omega_{m}\right], \end{array}\right.$$
we exclude from the spectrum the harmonics with one of the rotation frequencies and mesh frequency harmonics. Let us assume also that one of the teeth of the wheel gear is defective and the power of the stochastic modulation of the pinion gear harmonics is negligible. As a result of the foregoing assumptions, we can carry out the analysis of the filtered signal within the framework of the PCRP model:(13)$$\xi_{1}\left(t\right)=\sum_{k\in Z}\xi_{k0}\left(t\right)e^{ik\frac{2\pi }{P_{1}}t}.$$

The mean and covariance function of (13) are represented in Fourier series form:(14)$$m\left(t\right)=\sum_{k\in Z}m_{k}e^{i\Lambda_{k0}t}=m_{0}+\sum_{k\in N}\left[m_{k}^{c}\cos k\Lambda_{k0}t+m_{k}^{s}\sin k\Lambda_{k0}t\right]$$
(15)$$R\left(t,\tau \right)=\sum_{k\in Z}R_{k}\left(\tau \right)e^{i\Lambda_{k0}t}=R_{0}\left(\tau \right)+\sum_{k\in N}\left[R_{k}^{c}\left(\tau \right)\cos k\Lambda_{k0}t+R_{k}^{s}\left(\tau \right)\sin k\Lambda_{k0}t\right]$$
where $m_{k}=E\xi_{k0}\left(t\right)$, $R_{k}\left(\tau \right)=\sum_{p\in Z}r_{p-k,p}\left(\tau \right)e^{i\Lambda_{p0}\tau }$,
(16)$$r_{pq}\left(\tau \right)=E{\bar{\overset{\circ }{\xi }}}_{q0}\left(t\right)\xi_{p0}\left(t+\tau \right)$$
$m_{k}=\frac{1}{2}\left(m_{k}^{c}-im_{k}^{s}\right)$, and $R_{k}\left(\tau \right)=\left[R_{k}^{c}\left(\tau \right)-iR_{k}^{s}\left(\tau \right)\right]$. Modeling the real data, we must suppose that the Fourier series (13)–(15) is finite. The number of the highest harmonic can be obtained on the basis of preliminary processing of the data.

It follows from (16) that the zero^th^ covariance component is determined by the sum of the auto-covariance functions of modulating processes $\xi_{k0}\left(t\right)$. Thus, the time-averaged power of the signal $R_{0}\left(0\right)$ is equal to the sum of the modulation powers $r_{pp}\left(0\right)$:$$R_{0}\left(0\right)=\sum_{p=-L_{1}}^{L_{1}}r_{pp}\left(0\right)$$

The non-zero^th^ components $R_{k}\left(\tau \right)$ characterize the summary cross-correlations of the modulating processes, whose numbers differ by k. Consequently, we obtain the correlations of the spectrum components of frequencies shifted by values $\Lambda_{k0}$. These correlations are described by the spectral components:(17)$$f_{k}\left(\omega \right)=\frac{1}{2\pi }\int_{-\infty }^{\infty }R_{k}\left(\tau \right)e^{-i\omega \tau }d\tau$$

The values of the covariance components at the point τ=0 are their total characteristics in the time domain:(18)$$R_{k}\left(0\right)=\int_{-\infty }^{\infty }f_{k}\left(\omega \right)d\omega$$

The methodology for the vibration signal processing, developed in our investigations [10,17,28], has some specific content. It involves the sequential employment of the methods for the covariance and spectral analysis of the stationary random processes, the methods of searching for hidden periodicities of the first and the second order [29,30,31,32], their separation and individual analysis, empirical harmonic analysis on the basis of Fourier series and Fourier transformations, and the methods of PCRP covariance and spectral analysis [4,8,9,34,35,36]. The PCRP harmonic representation is used for the substitution of some techniques and for the interpretation of the processing results. Now, we detail briefly the main stages of the PCRP covariance analysis.

### 3.1. The Stationary Analysis

In order to study the general properties of the vibration signals, the estimators of the covariance function and of the power spectral density of PCRP stationary approximation are calculated in the initial stage:(19)$$\hat{R}\left(jh\right)=\frac{1}{K}\sum_{n=0}^{K-1}\left[\xi \left(nh\right)-\hat{m}\right]\left[\xi \left(\left(n+j\right)h\right)-\hat{m}\right],\;\hat{m}=\frac{1}{K}\sum_{n=0}^{K-1}\xi \left(nh\right)$$
(20)$$\hat{f}\left(\omega \right)=\frac{h}{2\pi }\sum_{n=-L}^{L}k\left(nh\right)\hat{R}\left(nh\right)\cos\omega nh$$

Here h=T/K is the sampling interval, j is the integer number, T is the realization length, K is the sample size, $L=\tau_{m}/h$ is some natural number, $\tau_{m}$ is the point of correlogram cutoff, and k(nh) is the covariance window. The analysis of the calculation results allow us to detect the presence of the deterministic component and to determine the relationship between the powers of the deterministic and stochastic oscillations to clarify the spectral composition of the raw signal.

### 3.2. The Detection and the Analysis of the Hidden Periodicities of the First Order

The estimator of the power spectral density, as a rule, includes continuous and discrete components. The latter are caused by deterministic oscillations. In order to estimate the period of these oscillations, the coherent, the component, and the least squares methods can be applied [9,10,17,29,30,31,32]. The values of the estimated period, apart from rare cases, are not multiples of the sampling interval and, therefore, the interpolation of the raw experimental data is required if the coherent technique is used. This causes additional processing error. The dependence of the processing results from the time reference point is a drawback of this approach as well. With this in mind, the component and the least squares (LS) methods are recommended for periodicity detection and for the estimation of the periods.

The LS statistics has the form:(21)$$F_{1}\left(\theta \right)=\frac{1}{2K+1}\sum_{n=-K}^{K}{\hat{m}}^{2}\left(\theta ,nh\right)$$
where
(22)$$\hat{m}\left(\theta ,nh\right)=\sum_{k=1}^{L_{1}}\left[{\hat{m}}_{k}^{c}\left(\theta \right)\cos k\frac{2\pi }{\theta }nh+{\hat{m}}_{k}^{s}\left(\theta \right)\sin k\frac{2\pi }{\theta }nh\right]$$
(23)$$\left\{\begin{array}{l} {\hat{m}}_{k}^{c}\left(\theta \right)\\ {\hat{m}}_{k}^{s}\left(\theta \right) \end{array}\right\}=\frac{2}{2K+1}\sum_{n=-K}^{K}\xi \left(nh\right)\left\{\begin{array}{l} \cos k\frac{2\pi }{\theta }nh\\ \sin k\frac{2\pi }{\theta }nh \end{array}\right\}$$
and θ is the so-called test period. The error caused by the aliasing effects of the first and the second kinds can be avoided if the sampling step h in (22) and (23) satisfies the inequalities [30,37]:(24)$$h\leq \frac{P_{1}}{2L_{1}+1},\;h\leq \frac{P_{1}}{2L_{2}+1}$$
where $L_{1}$ and $L_{2}$ are the numbers of the highest harmonics of the mean and covariance function, respectively. It should be noted that the values of the test period θ in (21) and (22) may be arbitrary and independent of the sampling step h.

The maximum values of (23) are close to the amplitudes of the individual harmonics of the mean function. The efficiency of the LS period estimator (21) is higher than the component estimator (23) efficiency because other harmonics of the amplitude spectrum are considered while processing the data. The functional (22), at the point of maximum $\theta ={\hat{P}}_{1}$, reaches a value close to the time-averaged power of the deterministic oscillations:$$Q_{d}=\frac{1}{2}\sum_{k=0}^{L}\left[{\left[{\hat{m}}_{k}^{c}\left({\hat{P}}_{1}\right)\right]}^{2}+{\left[{\hat{m}}_{k}^{s}\left({\hat{P}}_{1}\right)\right]}^{2}\right]$$

This quantity can exceed the amplitudes of the individual harmonics significantly. Therefore, it is advisable to use the functional (21) for the detection of low-power oscillations.

Knowing the period estimator value, we can form the component estimator [37,38] of the mean function:(25)$$\hat{m}\left(t\right)={\hat{m}}_{0}+\sum_{k=1}^{L_{1}}\left[{\hat{m}}_{k}^{c}\left({\hat{P}}_{1}\right)\cos k\frac{2\pi }{{\hat{P}}_{1}}t+{\hat{m}}_{k}^{s}\sin k\frac{2\pi }{{\hat{P}}_{1}}t\right]$$
where ${\hat{m}}_{0}=\hat{m}$. If conditions (24) are satisfied, relation (25) determines the mean function for all $t\in \left[0,\;{\hat{P}}_{1}\right]$ [37].

For the estimators of the amplitude and phase spectrum of the deterministic oscillations, we have:(26)$$\hat{A}\left(k\frac{2\pi }{{\hat{P}}_{1}}\right)=\sqrt{{\left[m_{k}^{c}\left({\hat{P}}_{1}\right)\right]}^{2}+{\left[m_{k}^{s}\left({\hat{P}}_{1}\right)\right]}^{2}},\;\hat{\varphi }\left(k\frac{2\pi }{{\hat{P}}_{1}}\right)=arctg\frac{m_{k}^{s}\left({\hat{P}}_{1}\right)}{m_{k}^{c}\left({\hat{P}}_{1}\right)},\;k=\bar{1,\;L_{1}}$$

It is reasonable to assume that the amplitudes of some harmonics increase as a gear fault develops. Therefore, for the quantitative evaluation of the gear condition we can use the aggregate amplitude:(27)$${\hat{A}}_{\Sigma }=\sum_{k=1}^{L_{1}}\hat{A}\left(k\frac{2\pi }{{\hat{P}}_{1}}\right)$$
or aggregate power of harmonics:(28)$${\hat{Q}}_{d}=\frac{1}{2}\sum_{k=1}^{L_{1}}{\hat{A}}^{2}\left(k\frac{2\pi }{{\hat{P}}_{1}}\right)$$

The change of the gear condition can be analyzed using the indicators defined by the ratios of the quantities
(29)$$I_{1}=\frac{{\hat{A}}_{\Sigma }^{\left(c\right)}}{{\hat{A}}_{\Sigma }^{\left(i\right)}},\;I_{2}=\frac{{\hat{Q}}_{d}^{\left(c\right)}}{{\hat{Q}}_{d}^{\left(i\right)}}$$
calculated for the initial and the current states.

### 3.3. The Analysis of the Hidden Periodicities of the Second Order

For the detection of the hidden periodicities of the second order, a formula similar to the one described above has to be applied. The LS functional of the covariance function has the form:(30)$$F_{2}\left(jh,\theta \right)=\frac{1}{2K+1}\sum_{n=-K}^{K}{\hat{R}}^{2}\left(nh,jh,\theta \right)$$
where
(31)$$\hat{R}\left(nh,jh,\theta \right)=\sum_{k=1}^{L_{2}}\left[{\hat{R}}_{k}^{c}\left(jh,\theta \right)\cos k\frac{2\pi }{\theta }nh+{\hat{R}}_{k}^{s}\left(jh,\theta \right)\sin k\frac{2\pi }{\theta }nh\right]$$
(32)$$\left\{\begin{array}{l} {\hat{R}}_{k}^{c}\left(jh,\theta \right)\\ {\hat{R}}_{k}^{s}\left(jh,\theta \right) \end{array}\right\}=\frac{2}{2K+1}\sum_{n=-K}^{K}\left[\xi \left(nh\right)-\hat{m}\left(nh\right)\right]\left[\xi \left(\left(n+j\right)h\right)-\hat{m}\left(\left(n+j\right)h\right)\right]\left\{\begin{array}{l} \cos k\frac{2\pi }{\theta }nh\\ \sin k\frac{2\pi }{\theta }nh \end{array}\right\}$$

The estimator of the covariance function period is found as the point of the maximum of the statistics (30) with respect to test period θ. The aliasing effects of the first and the second kinds are absent if the sampling step satisfies the inequality:(33)$$h\leq \frac{P_{1}}{4L_{2}+1}$$

At the point $\theta ={\hat{P}}_{1}$, the quantity (30) is close to the average value of time variation of the covariance function power for lag τ=jh:$$F_{2}\left(jh,\;{\hat{P}}_{1}\right)=\frac{1}{2}\sum_{k=1}^{L_{2}}\left[{\left[R_{k}^{c}\left(jh,{\hat{P}}_{1}\right)\right]}^{2}+{\left[R_{k}^{s}\left(jh,{\hat{P}}_{1}\right)\right]}^{2}\right]$$

To calculate the covariance function, the statistics
(34)$$\hat{R}\left(t,jh,{\hat{P}}_{1}\right)={\hat{R}}_{0}\left(jh\right)+\sum_{k=1}^{L_{2}}\left[R_{k}^{c}\left(jh,{\hat{P}}_{1}\right)\cos k\frac{2\pi }{{\hat{P}}_{1}}t+R_{k}^{s}\left(jh,{\hat{P}}_{1}\right)\sin k\frac{2\pi }{{\hat{P}}_{1}}t\right]$$
should be used. Here,
(35)$${\hat{R}}_{0}\left(jh\right)=\frac{1}{2K+1}\sum_{n=-K}^{K}\left[\xi \left(nh\right)-\hat{m}\left(nh\right)\right]\left[\xi \left(\left(n+j\right)h\right)-\hat{m}\left(\left(n+j\right)h\right)\right]$$

If condition (33) is satisfied, Equation (34) is an interpolation formula that allows us to calculate the covariance function values for all $t\in \left[0,{\hat{P}}_{1}\right]$ [37].

The estimator for variance $\hat{R}\left(t,0,{\hat{P}}_{1}\right)$ defines the time variations of the power of the vibration stochastic component and the estimator of the zero^th^ covariance component at the point j=0, i.e., a time-averaged value of this power. The quantities
(36)$$\hat{V}\left(k\frac{2\pi }{{\hat{P}}_{1}}\right)=\sqrt{{\left[R_{k}^{c}\left(0,{\hat{P}}_{1}\right)\right]}^{2}+{\left[R_{k}^{s}\left(0,{\hat{P}}_{1}\right)\right]}^{2}}\;\mathrm{and}\;\psi \left(k\frac{2\pi }{{\hat{P}}_{1}}\right)=arctg\frac{R_{k}^{s}\left(0,{\hat{P}}_{1}\right)}{R_{k}^{c}\left(0,{\hat{P}}_{1}\right)},\;k=\bar{1,\;L_{2}}$$
can be considered as the amplitude and phase spectrum of the power time variations, respectively.

As follows from the investigations already carried out [10,11,12,13,14,15,16,17,18,28], the vibration signal generated by a rotating machine acquires the properties of the periodical non-stationarity of the second order as the fault initiates. This means that the time variations of the variance and the covariance function estimators are the test indication of machine damage. Thus, indicators formed on the basis of covariance components are sensitive to the appearance of a fault. Such an indicator is determined by the ratio of the aggregate amplitude of the variance harmonics and the zero^th^ covariance component:(37)$$I_{3}=\frac{\sum_{k=1}^{L_{2}}\hat{V}\left(k\frac{2\pi }{{\hat{P}}_{1}}\right)}{{\hat{R}}_{0}\left(0\right)}$$

The accelerated growth of this indicator is evidence of the rapid deterioration of the mechanism properties.

## 4. The Analysis of the Natural Data

We consider below the results of the analysis of vibrations generated by a wind turbine gearbox (Figure 1) whose condition was monitored during the year. The position of the accelerometer mounted on the gearbox housing is marked with an arrow in Figure 1b.

The number of pinion gear teeth was 25 and, for the wheel, it equaled 94. The duration of the raw signal was 3.35 s (8192 samples). The vibration segments, which correspond to different stages of gear tooth failure, are shown in Figure 2. The speeds of the high-speed shaft (HSS) rotation were measured by means of a tachometer and for each stage were, respectively, 1451.55 rpm, 1442.85 rpm, and 1404.75 rpm. It can be seen from Figure 2 that for the second (Figure 2b) and the third (Figure 2c) cases, the raw signals include clear impacts caused by the presence of the evolutionary fault. The time intervals between the impacts are close to the period of pinion gear rotation.

### 4.1. The Stationary Approximation Properties

To ascertain the spectral composition of the vibration, the estimators of the spectral density for the signal stationary approximation were calculated using formulae (19) and (20). The Hamming window
$$k\left(\tau \right)=\left\{ \begin{array}{l} 0.54+0.46\cos\frac{\pi \tau }{\tau_{m}},\;\;\left|\tau \right|\leq \tau_{m},\\ 0,\;\;\left|\tau \right|>\tau_{m}, \end{array}\right.$$
where $\tau_{m}$ is a point of the correlogram cutoff, was used for the calculations. It follows from the results obtained (Figure 3), that the spectrum of vibration was located within the frequency range of 0...10 kHz (Figure 3a), but the dominant power portion belonged to the band limited by 3 kHz. The chart of the spectral density estimator for this frequency domain is shown in Figure 3b. The graphs on this chart have the form of a comb with different amplitudes and bandwidths. The estimator takes the peak values at the points, which coincide with the mesh frequency and its multiples, the pinion gear rotation frequency and its multiples, and their combinations. We also highlight the frequency bands that correspond to powerful resonances, i.e., $\left[f_{m},\;1.8f_{m}\right]$ and $\left[2.2f_{m},\;3f_{m}\right]$. The powers of the spectral components that correspond to wheel rotation (approximately 6.4 Hz) and its multiples are negligible. Hence, we can assume that the deterministic and the stochastic modulations, caused by PCRP oscillations of the input rotation period, are negligible too, and formally analyze the present data as a segment of the PCRP realization of the output period.

Further, we concentrated on the analysis of the signal properties at frequencies less than $1.8f_{m}$. Estimators of the covariance function and of the spectral density for the stationary approximation of the filtered signals corresponding to the three stages of the pinion tooth failure are given in Figure 4 and Figure 5.

The undamped tail is a distinctive feature of the covariance function estimators. As it follows from the covariance function of the PCRP stationary approximation,
(38)$$R\left(\tau \right)=R_{0}\left(\tau \right)+\frac{1}{2}\sum_{k=1}^{L_{1}}{\left|m_{k}\right|}^{2}\cos k\frac{2\pi }{P}\tau$$
the undamped tail includes cosine oscillations with amplitudes corresponding to the power of each deterministic harmonic. At the point τ=0, expression (38) defines an aggregate power of the deterministic and the stochastic oscillations. At the point $\tau_{r}=rP$, where r is a natural number for which $R_{0}\left(rP\right)\approx 0$, we obtain the value of the power of the deterministic oscillations. For the three stages of gear tooth degradation considered, the summary power of vibration was equal to $0.95G^{2}$, $5.84G^{2}$, and $7.73G^{2}$ and the power of the deterministic oscillations was $0.72G^{2}$, $5.12G^{2}$, and $6.73G^{2}$, respectively. It is evident that the part of stochastic oscillation power decreases as the fault grows. If, for the initial stage, this part is equal to 0.3, then for the last stage it equals only 0.14. Note that the undamped tail has a group structure. The time interval between the individual groups is close to the period of the output shaft rotation. Each group consists of seven–eight waves, so we can predict that the seventh–eighth rotation harmonics have the largest power. The presence of the undamped tail in the covariance function estimator induces the discrete components of the spectral density estimator, which are represented by the peaks at some frequencies (Figure 5). The detected peaks can also be the result of the narrow-band feature of the stochastic components. Thus, the mixed spectra obtained make it difficult to interpret the spectral estimation results and their quantitative analysis. We can only conclude that the time-averaged power of the vibration increases with the fault growth and, herewith, both the spectrum width increases and a new spectrum lines appear alongside an increase in their heights. As was noted above, it follows from Figure 4 that the time-averaged power of the stochastic part increases more slowly than power of the deterministic oscillations.

Proceeding from (19), (20) and (38) for the discrete spectrum estimator, we obtain
$${\hat{f}}_{d}\left(\omega \right)=\int_{-\infty }^{\infty }f_{d}\left(\omega_{1}\right)\lambda \left(\omega -\omega_{1}\right)d\omega_{1}$$
where
$$f_{d}\left(\omega \right)=\frac{1}{2}\sum_{k=1}^{L_{1}}{\left|m_{k}\right|}^{2}f\left(\omega -k\omega_{0}\right)$$
and
$$\lambda \left(\omega \right)=\frac{1}{2\pi }\int_{-\infty }^{\infty }k\left(\tau \right)e^{-i\omega \tau }d\tau$$

Hence,
$${\hat{f}}_{d}\left(\omega \right)=\frac{1}{2}\sum_{k=1}^{L_{1}}{\left|m_{k}\right|}^{2}\lambda \left(\omega -k\omega_{0}\right)$$

Since $\lambda \left(0\right)\leq \tau_{m}$, the peak values are not equal to the individual harmonic power and they change if the values of $\tau_{m}$ change. Therefore, the separation of the continuous and discrete components, and their individual analysis by means of adequate techniques, are required. In particular, this is important for the monitoring issue, since the discrete and continuous components can be caused by different faults.

### 4.2. Analysis of the Deterministic Oscillations

The estimation of the period is the initial issue of the separation and analysis of the deterministic oscillations of the vibration. Note that the accuracy of the period estimation must be relatively high to reach the minimal displacement of the initial point of the averaging. For example, if the period estimation error δP is equal to 0.01P, then, after the synchronous averaging of the realization of the T=100P length, this displacement is equal to period value P. It is obvious that such displacement is undesirable, therefore, it is necessary to carry out the period estimation and the following calculation of the harmonic amplitude on the basis of the analyzed realization, and to use formulae whose extreme values simultaneously provide the extremes of the formulae for the estimation of the corresponding signal parameters. The least squares method was applied to the period estimation since, in this case, we can consider the aggregate power of the chosen harmonics of the deterministic parts, which clearly increased the estimation efficiency. Note that the systematic error of the period least squares estimators has the order $O\left(T^{-2}\right)$ and the mean-square one, $O\left(T^{-\frac{3}{2}}\right)$ [30,31].

The charts of the dependence of the square functional (29) on the test frequency f=1/θ for the three stages of gear failure are shown in Figure 6. These were calculated on the basis of Formula (22) for $k=\bar{5,\;12}$. The points of the functional maximum for each of the considered stages, with an accuracy of up to three digits after the comma, correspond to the basic frequency estimator and are equal to ${\hat{f}}_{0}=\frac{1}{{\hat{P}}_{1}}=24.206\;\mathrm{Hz}$ (Figure 6a), ${\hat{f}}_{0}=24.055\;\mathrm{Hz}$ (Figure 6b), and ${\hat{f}}_{0}=23.423\;\mathrm{Hz}$ (Figure 6c). The estimated values of the basic frequency of the deterministic oscillations were very close to the values provided by the tachometer measurements, namely 24.192 Hz, 24.047 Hz, and 23.412 Hz.

Proceeding from the estimated basic frequency values, the harmonic amplitudes were calculated on the basis of expressions (23) and (26). The amplitude spectra of the vibration deterministic part are represented in the form of diagrams in Figure 7 and the harmonic amplitude values $\hat{A}\left(k{\hat{f}}_{0}\right)$ are provided in Table 1.

The first harmonics of the deterministic part spectra can be interpreted as the order harmonics of the shaft rotation frequency; the twenty-fifth harmonic corresponds to the first harmonic of the mesh frequency, and the frequencies of the higher harmonics are linear combinations of the mesh and rotation frequencies. In the first stage, the amplitude of the mesh frequency harmonic is the largest.

As the fault grows, the harmonics of the 6^th^–9^th^ orders become dominant, while the general form of the amplitude spectra remains similar. The aggregate amplitudes of the harmonics ${\hat{A}}_{\Sigma }$ (27) for each stage are, respectively, equal to 3.47, 7.44, and 10.50, while the aggregate harmonic powers (28) are 0.36, 3.52, and 4.63. The indicator $I_{1}$ (38) changed in the following way: $I_{1}=2.14$, 3.03, while the indicator $I_{2}$ (29) changed to $I_{2}=9.78$, 12.86. On the basis of the sine and cosine Fourier coefficients (23) using the interpolation Formula (25), the PCRP mean function can be calculated for all $t\in \left[0,\;{\hat{P}}_{1}\right]$ (Figure 8).

Proceeding from [32,36], and taking into consideration the values of the covariance function calculated below, we can conclude that, for the given realization length, the standard deviation of the mean function estimator $\sigma \left[\hat{m}\left(t\right)\right]$ is smaller than 0.01. As was expected, the deterministic oscillations had a group structure. The time intervals between the groups were close to the period of shaft rotation and each group consisted of approximately eight waves.

### 4.3. Analysis of the Stochastic Oscillations

The further analysis of the gearbox conditions was conducted on the basis of the vibration residues obtained by means of the centering of the raw signals on the estimator of the PCRP mean function, i.e., $\overset{\circ }{\xi }\left(t\right)=\xi \left(t\right)-\hat{m}\left(t\right)$. The graphs of the covariance function and the spectral density estimators of the vibration residuals are given in Figure 9, Figure 10 and Figure 11. The covariance function estimators had the form of slowly damped groups following one after the other over the rotation period. These groups became clearly observable for the second (Figure 9b) and the third (Figure 9c) stages. As the lag increased, the estimators decayed to low-power fluctuations, so we concluded that the deterministic oscillations were fully extracted from the vibration signal. Thus, the spectral densities of the vibration residuals include only the continuous parts (Figure 10 and Figure 11). The comb-like forms of the spectral densities estimators indicated a narrow-band modulation of the PCRP carrier harmonics of both the low- and high-frequency range. This means that the modulating processes can be represented in the form of the sum of the low- and high-frequency narrow-band components. These components can be modeled as respective Rice representations [39,40,41,42,43]. Conclusions about the correlations, or lack of correlations, between these components within the low- and high-frequency domains can be drawn only on the basis of the results of a PCRP analysis. The detection of the periodic time variation of the vibration residual variance is the first test procedure of this approach. The detection was carried out using statistics (30) and (31) for $k=\bar{1,\;9}$.

For each stage of gear tooth failure, the graph of the functional dependence on test frequency 1/θ includes a clearly defined peak (Figure 12) at the point to be considered as an estimator of the variance period or basic frequency. For each case, the estimated values of the basic frequency are $f_{0}=1/{\hat{P}}_{1}=24.196\;\mathrm{Hz}$, 24.075 Hz, and 23.423 Hz.

These values differ only insignificantly from the basic frequency estimators of the vibration mean functions. The clearly defined peak on the graph in Figure 12a corresponds to the early stage of the fault initiation. Considering the powers of the peaks in Figure 12b,c, we concluded that a defect had developed.

The quantitative appraisal of the gear pair condition is given below.

Knowing the values of ${\hat{f}}_{0}$, on the basis of expressions (32) and (36), we calculated the Fourier coefficients for the variance and the amplitude spectrum of the variance time variation, which are represented in the form of the diagrams in Figure 13. As can be seen, for the first stage harmonic amplitude $\hat{V}\left(k{\hat{f}}_{0}\right)$, k≠0 were negligible. For the rest stages, the amplitude spectrum slowly decayed as the frequency increased. The rate of decay decreased within the frequency range 54–145 Hz and it was a feature of both stages. The amplitudes for k>15 were negligible. Then, it follows from (17), that the correlations of the spectrum components, the frequency interval between which is greater than 350 Hz, were weakly correlated. Thus, we concluded that the low-frequency and high-frequency modulations were non-correlated. This means that the variance Fourier coefficients are determined by the correlation of the narrow-band components separately in low-frequency and high-frequency domains. The first coefficient is determined by the correlations of narrow-band components shifted by ${\hat{f}}_{0}$, while the second coefficient is shifted by $2{\hat{f}}_{0}$, etc. Since the bandwidth Ω for the correlated components is limited, the number of the possible correlations decreases as the coefficient numbers increases. The harmonic amplitudes decrease too. In the considered case, Ω≤350 Hz, therefore, the number of the significant variance harmonics $L_{2}$ could not be larger than 15. As can be seen, this inference was confirmed by processing results.

As was noted above, at the first stage, the covariance components, except for the zero^th^, were insignificant; however, it follows from Figure 12a that we cannot ignore the features of the newly appeared second order non-stationarity. This indicates the initiation of a local fault caused by parallel misalignment detected by further maintenance. We should note that the damage was detected using LS statistics, the extreme value of which is determined by the sum of the time-averaged powers of all possible variance harmonics with non-zero amplitudes. Using the component statistics (32) is not efficient for the discovering of hidden periodicities on the basis of realizations acquired in the early stages. In Figure 14, the graphs of statistics
$$\left|{\hat{R}}_{k}\left(0,\theta \right)\right|={\left[{\left[R_{k}^{c}\left(0,\theta \right)\right]}^{2}+{\left[R_{k}^{s}\left(0,\theta \right)\right]}^{2}\right]}^{\frac{1}{2}}$$
with the maximum values (Figure 13 and Table 2) are presented. As can be seen, it is difficult to make any inference from these charts because the revealed extreme values are very small and the frequency dependences are similar to chaotic oscillations.

The values of the harmonic amplitudes are given in Table 2. We should note that the values of the variance Fourier coefficients cannot exceed the value of the zero^th^ coefficient. This fact follows from the PCRP harmonic representation (13), namely from relation (16):$$\left|R_{k}\left(\tau \right)\right|\leq \sum_{p\in Z}r_{p-k,p}\left(0\right)\leq \sum_{p\in Z}r_{pp}\left(0\right)=R_{0}\left(0\right)$$

The values of the indicator $I_{3}$ (37) for each stage of the failure were: 1.29, 5.15, and 6.13. As we can see from Table 2, it follows that the time-averaged power of the vibration $R_{0}\left(0\right)=\int_{-\infty }^{\infty }f\left(\omega \right)d\omega$ also increased as the fault grew. The ratio of the current value of the zero^th^ covariance component $R_{0}^{\left(c\right)}\left(0\right)$ and the initial $R_{0}^{\left(i\right)}\left(0\right)$ component for the second stage was equal to 3.23, and, for the third stage, was equal to 4.44. To take this feature into account, we form the indicator
$$I_{4}=\frac{\Delta {\hat{R}}_{0}\left(0\right)+\sum_{k=1}^{L_{2}}\hat{V}\left(k{\hat{f}}_{0}\right)}{{\hat{R}}_{0}^{\left(i\right)}\left(0\right)},$$
where $\Delta {\hat{R}}_{0}\left(0\right)={\hat{R}}_{0}^{c}\left(0\right)-{\hat{R}}_{0}^{\left(i\right)}\left(0\right)$. For the indicator $I_{4}$, we accordingly have the following values: 1.29, 13.82, and 30.72. The significant increase of the indicator $I_{4}$ demonstrates its high sensitivity to the changes of gear conditions in spite of the small values of the variance Fourier coefficients in comparison with the amplitudes of the harmonics of the deterministic oscillations.

The indicators for the fault detection formed on the basis of the mean and the variance spectrum were also proposed in other works [7,12,44,45]. They have the forms of so-called indicators of the cyclostationarity. However, it was assumed that the values of the cyclic frequency are known, as the corresponding amplitudes were calculated on the basis of the experimental data. These values were determined proceeding from the given technical parameters of the rotating machine units. Since these parameters change during machine operations, the chosen frequency value can differ from the real value. In this case, the processing results could be, essentially, distorted. Therefore, the estimation of the cyclic frequencies on the basis of the given realization is the first issue of the practical vibration analysis.

Note that the indicators used in the paper differ from the indicators of cyclostationarity. The gear state is described by the ratio of the power of the time-changing of the mean or the variance to the initial values of these quantities, but not to the time-averaged variance for each state. The latter essentially changes with the fault growth. Therefore, it is advisable to take into consideration these changes, as was done above.

The charts of the variance time changes obtained on the basis of interpolation Formula (15) are shown in Figure 15. In the time interval equal to a period of non-stationarity, these changes included a significant impact caused by the faulty gear interaction. The impacts were especially powerful for the last cases when the tooth defect was well developed and close to breakage.

The pinion tooth breakage was confirmed by the site team after the borescope inspection of the gearbox parallel stage was performed (Figure 16). Calculating the relative standard deviation of the covariance function estimation $\sigma_{r}\left[\hat{R}\left(t,0\right)\right]$ using the formulae obtained in [33,38], we concluded that for the given realization length it was smaller than 0.04.

The time changes of the variance can be visually demonstrated by involuting its time dependence on the plane X0Y, forming the plot of a closed central curve, the coordinates of which are determined by the equations:$$x\left(t\right)=\hat{R}\left(0,t\right)\cos\left(2\pi t/{\hat{P}}_{1}\right)$$
$$y\left(t\right)=\hat{R}\left(0,t\right)\sin\left(2\pi t/{\hat{P}}_{1}\right)$$

In the case of the incipient fault, the curve chart has a form close to a circle, the radius of which is defined by the variance of the vibration of the stochastic part. If the fault grows, the curve loses its central symmetry and acquires an asymmetric form, which characterizes the damage. The charts of these curves obtained for the estimators of the vibration stochastic part variance for the different stages of the gear tooth breakage are given in Figure 17. As can be seen from the charts, the curve form visually represents the changes of the gear conditions of operations. The loss of symmetry was already visible at the first stage. For the rest stages, the curves were lengthened in the same direction that testifies to the localization of the fault. In our opinion, this chart representation makes the observation of mechanism condition easier.

The specific features of the fault can also be established on the basis of the analysis of the covariance components on the time lag. The graphs of covariance components dependences on the time lag are given in Figure 18 and Figure 19. The charts of these quantities for large lags have the form of damping groups (Figure 20), as for the zero^th^ component (Figure 9).

In the considered case, the cosine covariance components values were dominant; therefore, only their plots are shown in Figure 19 and Figure 20. The established group structures of the covariance components are, according to their general representation (13), if it is assumed that modulations are described by the high-frequency narrow-band jointly stationary random processes.

In this case, the modulating random processes can be represented in the form:(39)$$\xi_{k_{0}}\left(t\right)={µ}_{k}\left(t\right)e^{i\lambda_{0}t}+\nu_{k}\left(t\right)e^{i\lambda_{0}t}$$
where $\lambda_{0}$ is the gear pair resonant frequency, and ${µ}_{k}\left(t\right)$ and $\nu_{k}\left(t\right)$ are non-correlated stationary random processes. For the auto- and cross-covariance functions of (39), we have:$$r_{kk}\left(\tau \right)=r_{kk}^{\left(µ\right)}\left(\tau \right)e^{i\lambda_{0}\tau }+r_{kk}^{\left(\nu \right)}\left(\tau \right)e^{-i\lambda_{0}\tau }\;r_{kl}\left(\tau \right)=r_{kl}^{\left(µ\right)}\left(\tau \right)e^{i\lambda_{0}\tau }+r_{kl}^{\left(\nu \right)}\left(\tau \right)e^{-i\lambda_{0}\tau }$$
where $r_{kl}^{\left(µ\right)}\left(\tau \right)=E{\bar{µ}}_{k}\left(t\right){µ}_{l}\left(t+\tau \right)$ and $r_{kl}^{\left(\nu \right)}\left(\tau \right)=E{\bar{\nu }}_{k}\left(t\right)\nu_{l}\left(t+\tau \right)$. Then, the zero^th^ covariance component
$$R_{0}\left(\tau \right)=\overset{15}{\underset{l=-15}{\sum }}\left[r_{ll}^{\left(µ\right)}e^{i\lambda_{0}\tau }+r_{ll}^{\left(\nu \right)}\left(\tau \right)e^{-i\lambda_{0}\tau }\right]e^{il\omega_{0}\tau }$$
and the non-zero^th^ components
$$R_{k}\left(\tau \right)=\overset{L_{k}}{\underset{l=-L_{k}}{\sum }}\left[r_{l-k,l}^{\left(µ\right)}e^{i\lambda_{0}\tau }+r_{l-k,l}^{\left(\nu \right)}\left(\tau \right)e^{-i\lambda_{0}\tau }\right]e^{il\omega_{0}\tau }$$
where $L_{k}$ are the numbers of the correlated components, which are determined by superposition of decaying harmonics with frequencies $\lambda_{0}\pm k\omega_{0}$. Since the item frequencies are closed, these superpositions have group structures. The differences between the frequencies are equal to $k\omega_{0}$; therefore, the time intervals between groups are close to the rotation period. In Figure 20, we can distinctly see more then twelve groups. Thus, the vanishing interval of correlation for the stochastic oscillations considerably exceeded the rotation period. Such covariance structure causes a comb-like form of the spectral density. To specify the modulation properties, the band-pass filtration and the Hilbert transformation were applied [39,40,41,42].

## 5. Discussions

The techniques of vibration PCRP analysis proposed in [10,17,33] for early fault detection differ from the techniques for so-called cyclostationary analysis that are traditionally employed in the literature [13,14,15,46,47,48,49,50]. In Figure 21, the main stages of both approaches are shown for comparison.

The cyclostationary analysis involves the calculation of the cyclic auto-correlation function, depending on time and lag, and its two-dimensional Fourier transforms, which includes the search for correlated harmonics, the calculation of coherence functions, and their integrating, and the search for informative frequency band, using the various procedures [51,52,53,54,55,56,57,58,59,60,61,62], etc.

The PCRP analysis was provided in the time-frequency domain without transition into the dual-frequency domain. The time structure of the vibration signal was investigated by decomposition of the first and the second order moment functions into Fourier series. The amplitude spectra of the vibration deterministic component and time variations power for the stochastic part were used to describe the machinery state. The stationary analysis was carried out to ascertain the general properties of the vibration spectral composition and to determine the frequency interval for the discovery of hidden periodicities.

The effective techniques for discovering hidden periodicities of the first and second order, developed in [9,34,35,36,37], provide the period for the deterministic oscillations and the time variation of the moment functions of the second order for each individual realization with the required accuracy. It enables the estimation of the respective amplitude spectra, which can be used as a basis for assessing machinery conditions. The variance amplitude spectrum is defined by the modulus of the covariance components (cyclic functions) at the point τ=0:$$\left|R_{k}\left(0\right)\right|=\int_{-\infty }^{\infty }f_{k}\left(\omega \right)d\omega ,\;k=\bar{1,L_{2}}$$

The amplitude of the individual harmonics for order k is the total characteristic for the correlations of the spectral harmonics whose frequencies are shifted by $k\omega_{0}$. These quantities are complex; therefore, they cannot be called the cyclic power spectrum. The phase spectrum
$$\varphi_{k}\left(0\right)=arctg\frac{R_{k}^{s}\left(0\right)}{R_{k}^{c}\left(0\right)},\;k=\bar{1,L_{2}}$$
can also be used to characterize the variance time variations.

Summarizing the amplitudes of all the numbers, we obtained the total characteristic for all possible correlations of the spectral harmonics for stochastic vibrations, although the analysis was carried out only in the cyclic frequency domain within the framework of the Fourier series harmonic analysis.

The time-averaged power of stochastic oscillations, which was determined by $R_{0}\left(0\right)$, increased as the fault grew; this motivated the involvement of the increment $\Delta R_{0}\left(0\right)$ in the formula for the fault detection indicator. Thus, we expected that the indicator $I_{4}$, formed on the basis of all variance Fourier coefficients, would be as sensitive as possible to changes in the gear pair conditions.

It follows from (18) that the variance time variations in general are not localized in the frequency domain. The maximum frequency distance between correlated harmonics is determined by the highest number of the variance harmonic $L_{2}$ and is equal to $L_{2}\omega_{0}$. This means that the bandwidth for the filtering of the raw signal cannot be narrower that $L_{2}\omega_{0}$ and must be carried out over the whole signal frequency band. If these conditions are not fulfilled, the filtering results in the decrease of both amplitudes for variance harmonics and also their number. These vibration non-stationarity properties must be considered when the so-called informative frequency band is selected.

Comparing the cyclostationary and PCRP analyses, we can consider the so-called “envelope” (“high-resonance”) analysis devised in [63], which was widely used [14,15,46,48]. The envelope analysis typically consists of high-frequency band-pass filtering around some resonance frequency and of constructing the analytic signal ζ(t)=ξ(t)+iη(t), where η(t) is Hilbert transform for ξ(t) and the envelope $\mu \left(t\right)={\left[\xi^{2}\left(t\right)+\eta^{2}\left(t\right)\right]}^{\frac{1}{2}}$, and also the Fourier transform of the envelope. For many years, the envelope spectrum was recognized as one of most effective diagnostic tools for rotary machines [48,50]. At present, the square envelope spectrum is seen as preferable [64,65].

The envelope analysis was devised as an empirical technique [65,66,67,68]. It has to be applied to a purely random part of the signal; therefore, the deterministic components must be extracted. It is said that the modulus of the analytic signal is a low-frequency deterministic function describing the signal envelope [50,60]. However, theoretical analysis of the Hilbert transform of PCRP as a vibration model and the corresponding analytic signal show that this judgment is incorrect [39,40,41,42,43]. In fact, such an envelope does not exist [43,46,50]. The sum of squares of the accordingly filtered signal and its Hilbert transform is not a low-frequency deterministic function describing the squared envelope. On the contrary, it is the pure stochastic high-frequency random signal. The mathematical expectation of this signal is equal to the variance of the analytic signal. This variance is a function that is periodic in time, and its Fourier coefficients are determined by [43]:(40)$$B_{k}^{\left(\xi \right)}\left(0\right)=2\left[\int_{-\infty }^{\infty }f_{k}\left(\omega \right)d\omega -\int_{0}^{k\omega_{0}}f_{k}\left(\omega \right)d\omega \right]$$

From this formula, it follows that, for the high-frequency modulation, when $f_{k}\left(\omega \right)\neq 0$ only for $\omega \notin \left[0,k\omega_{0}\right]$, the quantity (40) is equal to the doubled signal covariance component. This means that the signal variance and the variance of its Hilbert transform are the same. Thus, the envelope analysis techniques are not the demodulating procedures; they cannot yield new results if they are compared to the analysis of raw signal variance. Consequently, it is advisable to use PCRP techniques to search for hidden periodicities in this virtual “square envelope”. The Fourier transform is not an applicable procedure in this case, and its results are not consistent. The use of PCRP techniques is more direct, and it is the more effective method for early fault detection.

It should be noted that the variance of the cyclic statistics, which is used in the “square envelope” analysis, has an order $O\left(T^{-1}\right)$, while the variance of the basic frequency estimator has an order $O\left(T^{-3}\right)$, and LS estimation provides the essentially greater SNR (signal-to-noise ratio). Since the modulus of non-zero^th^ covariance components $\left|R_{k}\left(0\right)\right|$ is always smaller than $R_{0}\left(0\right)$, i.e., $\left|R_{k}\left(0\right)\right|\leq R_{0}\left(0\right)$ and $\forall \;k=\bar{1,L_{2}}$, LS estimation has an evident advantage in search of hidden periodicities.

For a known basic frequency, the cyclic (component) estimation can be considered as filtration with a transfer function in the form of a comb, reaching the peaks at points $f=k{\hat{f}}_{0}$ and $\forall \;k=\bar{1,L_{2}}$. These peaks become sharper as the realization length increases. This approach allows us to increase the processing accuracy and to avoid the laborious procedures that are usually used to improve traditional techniques based on the discrete Fourier transform (see, for example, [50,51]).

The amplitude spectrum of the deterministic oscillations and, most of all, the amplitude spectrum of the time variations of the stochastic vibration power, characterize the fault features. The indicators formed on the basis of these spectra can be efficiently used for the analysis of machinery conditions. The greatest sensitivity of the indicator $I_{4}$ to the changes of the mechanism state is explained by both increase of the time averaged power for the stochastic oscillations $R_{0}\left(0\right)=\overset{\infty }{\underset{-\infty }{\int }}f_{0}\left(\omega \right)d\omega$ and also the correlations of the spectrum components, which are determined by the variance Fourier coefficients $R_{k}\left(0\right)=\overset{\infty }{\underset{-\infty }{\int }}f_{k}\left(\omega \right)d\omega$. As noted above that, relative change of the time averaged power for the second stage was equal to 3.23 and, for the third stage, was equal to 4.43. The indicator $I_{3}$, which is determined by the quantities $R_{k}\left(0\right)$ and k≠0, was equal, accordingly, to 5.15 and 6.13, while the total indicator $I_{4}$ was equal to 13.82 and 30.72, i.e., its sensitivity is the highest. Proceeding from the numerical results of the processing of numerous time series for the vibration of a wind turbine gearbox, we can outline some stages of fault development (Table 3). We should note that the emergency stage of the development of a fault is characterized by the rapid increase of both indicators. We recommend applying these indicators in practice. Note that the indicators’ numerical values are obtained on the basis of signal analysis in the frequency range of up to 1 kHz.

## 6. Conclusions

A model in the form of the BPCRP was proposed in this paper for the analysis of the vibrations of a damaged gear pair. The interaction of the deterministic oscillations of the two wheels was characterized by the BPCRP mean function and the interaction of the stochastic oscillations by the BPCRP covariance function. The Fourier series of the mean and covariance function consisted of the harmonics of the wheels’ rotation frequencies, their multiples, and combinations. The concrete harmonic compositions of the deterministic and the stochastic oscillations depend on the degree of the fault development and its location.

It was shown that the simpler stochastic models for gear pair vibration used in the literature are particular cases of the BPCRP stochastic series representation.

The PCRP approach was used in this paper to analyze the vibration of a wind turbine gearbox. It showed that the first and the second order PCRP parameters of the vibration at the frequency band $\left[0,1.8f_{m}\right]$, where $f_{m}$ is the mesh frequency, are sufficiently sensitive to state change, and they provided, to the full extent, the successful detection of the fault and monitoring of its development.

It was established that the mean function LS statistics for the period estimation had sharp peaks for all the analyzed stages. This result proved that powerful deterministic oscillations existed in the vibration structure. Determining the maximum point to an accuracy of three decimal places was accepted for the period estimators and, using their values, the amplitude of each harmonic was calculated and the amplitude spectrum was obtained. The spectrum forms for all stages differed insignificantly and its width covered practically the whole of the investigated frequency band. The low-frequency harmonics could be interpreted as order harmonics of the rotation frequency, the twenty-fifth harmonic was the first mesh-frequency harmonic and the frequencies of the higher harmonics were linear combinations of the mesh and the rotation frequencies. If, for the first stage, the amplitude of the mesh-frequency harmonic is dominant, then for the other stages the amplitudes of the 6th–9th order harmonics are the largest. The summary power of the harmonic rapidly increased as the fault grew. For the last stage, the power of the deterministic oscillations was more than nine times its value for the first stage. Using the interpolated component statistics, the mean function was calculated for all t∈[0,P] and the deterministic and the stochastic oscillations were divided.

The LS functional was employed for discovering the hidden periodicities of the second order. Its dependences on the test period had sharp peaks at the points that were accepted as the periods of the variance time changing. The availability of such growth peaks was evidence that a local fault had occurred and was developing.

On the basis of the period estimator, the variance Fourier coefficients were calculated and variance amplitude spectrum was formed. This spectrum characterizes the time periodic variations of the stochastic oscillation power. The power periodic variations are test features for the local fault detection. The amplitude of the individual harmonic order k is the total characteristic of the correlation of the spectral component whose frequencies are shifted by $k\omega_{0}$.

The summary amplitude of the variance harmonics was chosen for the comparison of the different states. The values of amplitudes for harmonics whose order is larger than twelve are negligible. This means that the spectral components, the frequency intervals between which are greater than 280 Hz, are weakly correlated. Thus, we concluded that low-frequency and high-frequency modulations are non-correlated.

Time variations of the variance do not occur if the fault is absent, thus it is advisable to choose, for the quantitative characterization of the state change, an initial value of the zero^th^ covariance component $R_{0}\left(0\right)$, which determines the average power of the stochastic oscillations. The average power increases as the fault grows, because this increase is advisable to be included to the formula for the diagnostic indicator. It is shown that the change of this indicator considerably exceeds the change of the deterministic indicator that is defined by the power of the deterministic vibrations, while the power of the latter is considerably larger than the power of the stochastic vibrations. The results obtained give grounds to recommend the stochastic power indicator for the monitoring of wind turbine gearboxes.

## Figures and Tables

**Figure 1 sensors-21-06138-f001:**
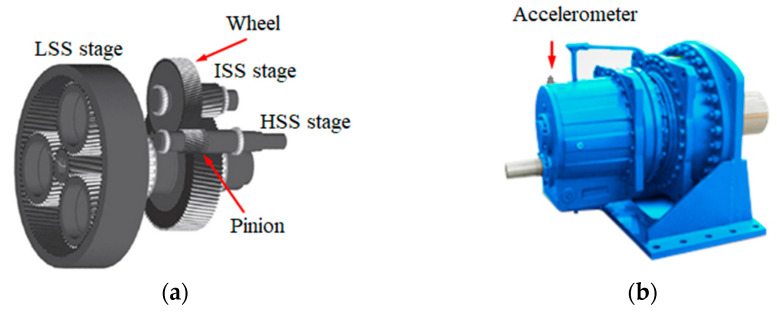
Schematic (**a**) and general (**b**) view of the WTG gearbox.

**Figure 2 sensors-21-06138-f002:**
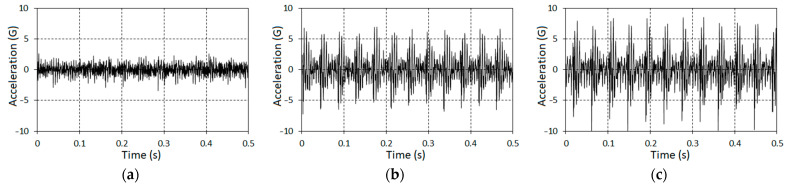
The segments of vibration realizations for three (**a**–**c**) stages of tooth failure.

**Figure 3 sensors-21-06138-f003:**
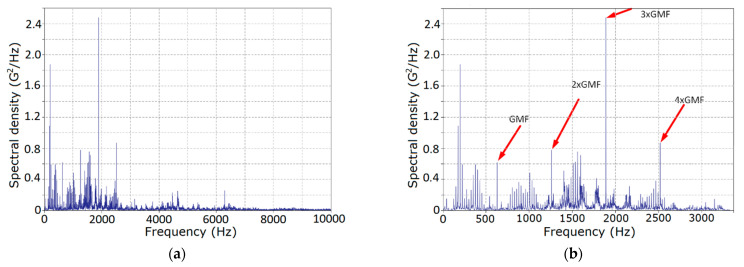
The estimators of the power spectral densities of the stationary approximation for the raw signal: (**a**) original and (**b**) zoomed.

**Figure 4 sensors-21-06138-f004:**
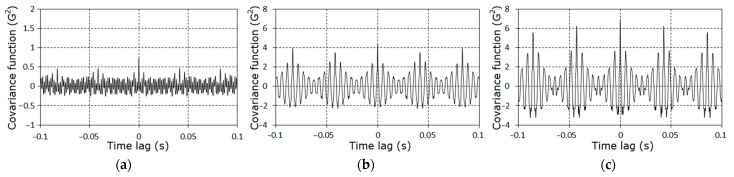
The covariance function estimators for the filtered signals corresponding to the first (**a**), second (**b**), and third (**c**) stages of the gear tooth failure.

**Figure 5 sensors-21-06138-f005:**
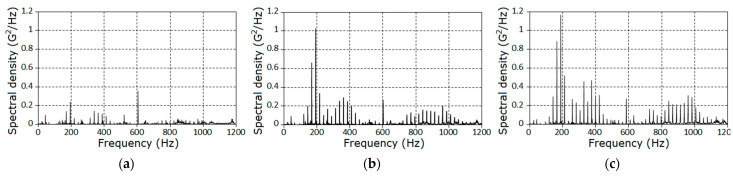
The power spectral density estimators for the filtered signals for each stage (**a**–**c**).

**Figure 6 sensors-21-06138-f006:**
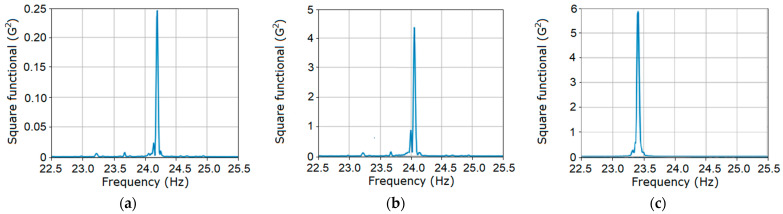
The dependence of the square functional of the first order on the test frequency for the first (**a**), the second (**b**) and the third (**c**) stages.

**Figure 7 sensors-21-06138-f007:**
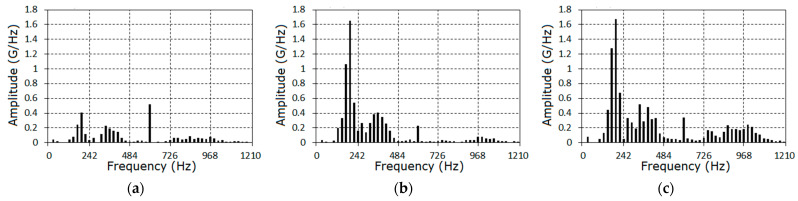
The amplitude spectra of the deterministic oscillations for the first (**a**), the second (**b**) and the third (**c**) stages.

**Figure 8 sensors-21-06138-f008:**
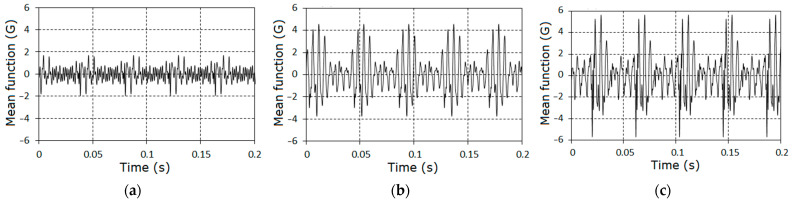
The mean function estimators of the vibration for the three stages (**a**–**c**) of gear tooth failure.

**Figure 9 sensors-21-06138-f009:**
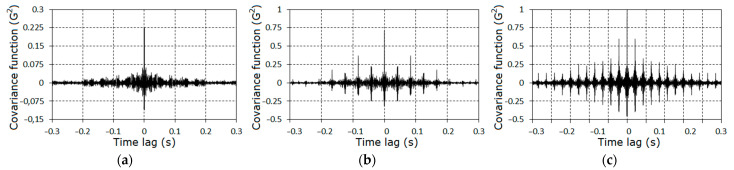
The estimators of the covariance function of the vibration of stochastic parts for the first (**a**), the second (**b**) and the third (**c**) stages.

**Figure 10 sensors-21-06138-f010:**
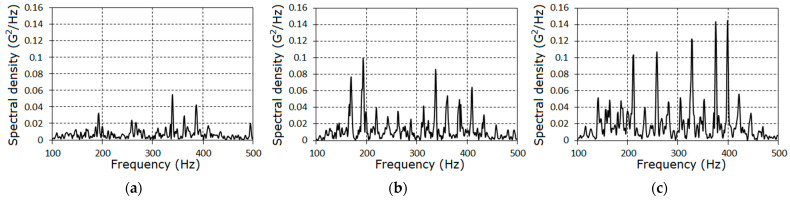
The estimators of the spectral densities of the vibration of stochastic parts in the low-frequency domain for the first (**a**), the second (**b**) and the third (**c**) stages.

**Figure 11 sensors-21-06138-f011:**
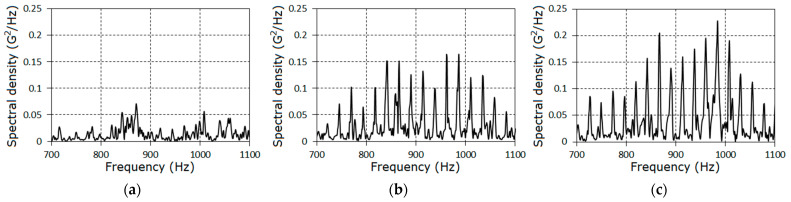
The estimators of the spectral densities of the vibration of stochastic parts in the high-frequency domain for the first (**a**), the second (**b**) and the third (**c**) stages.

**Figure 12 sensors-21-06138-f012:**
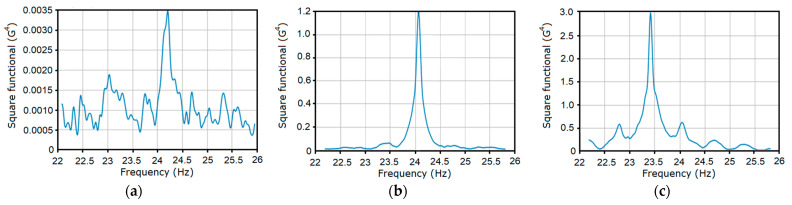
The dependences of the second order square functional (38) on the test period for the first (**a**), the second (**b**) and the third (**c**) stages.

**Figure 13 sensors-21-06138-f013:**
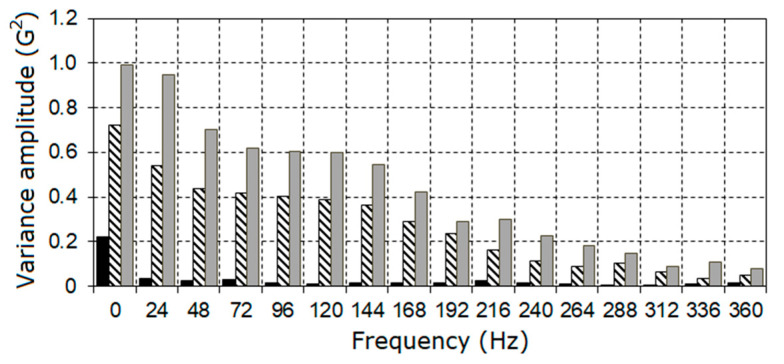
The amplitude spectrum of the variance periodical variation.

**Figure 14 sensors-21-06138-f014:**
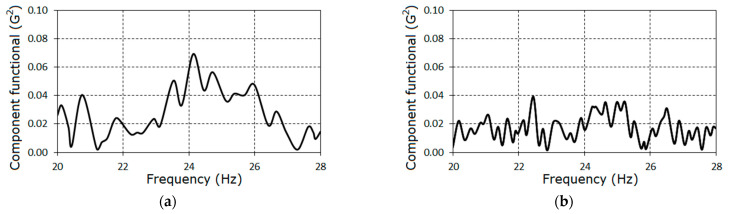
The first (**a**) and the second (**b**) component variance functionals for the first stage.

**Figure 15 sensors-21-06138-f015:**
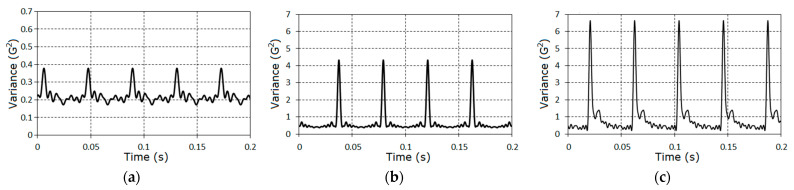
The estimators of the variance for the vibrations stochastic parts for the first (**a**), the second (**b**) and the third (**c**) stages.

**Figure 16 sensors-21-06138-f016:**
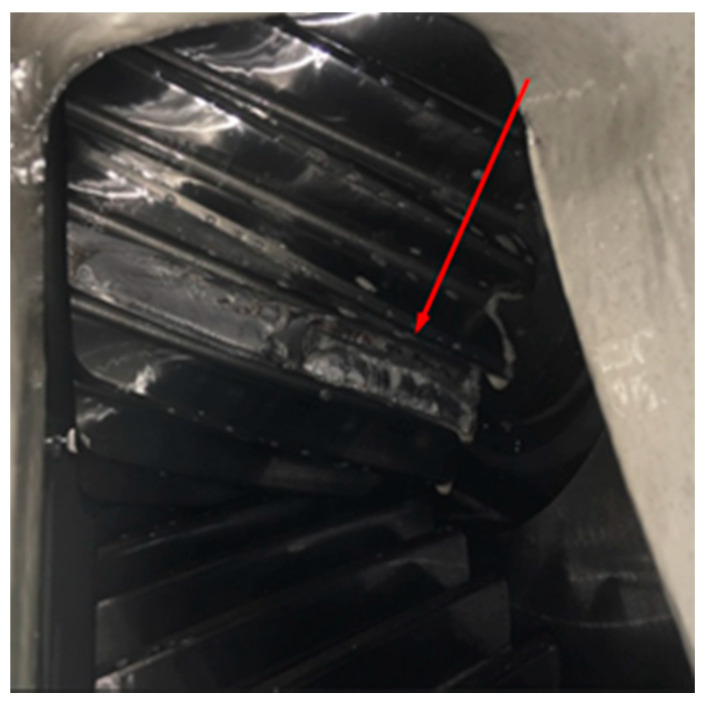
The liberated tooth of the pinion gear at gearbox parallel stage.

**Figure 17 sensors-21-06138-f017:**
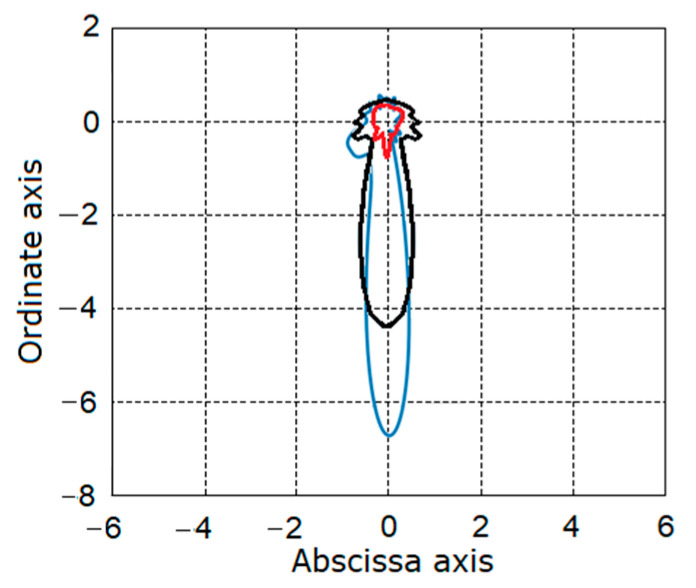
The time changes of the variance estimators on the plane.

**Figure 18 sensors-21-06138-f018:**
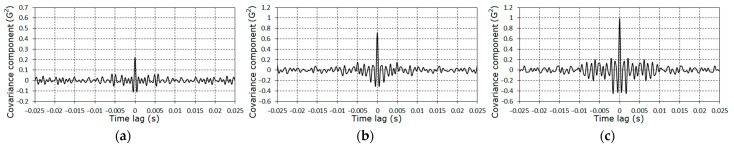
The dependences of the zero^th^ covariance component estimators on the time lag for the first (**a**), second (**b**), and third (**c**) stages.

**Figure 19 sensors-21-06138-f019:**
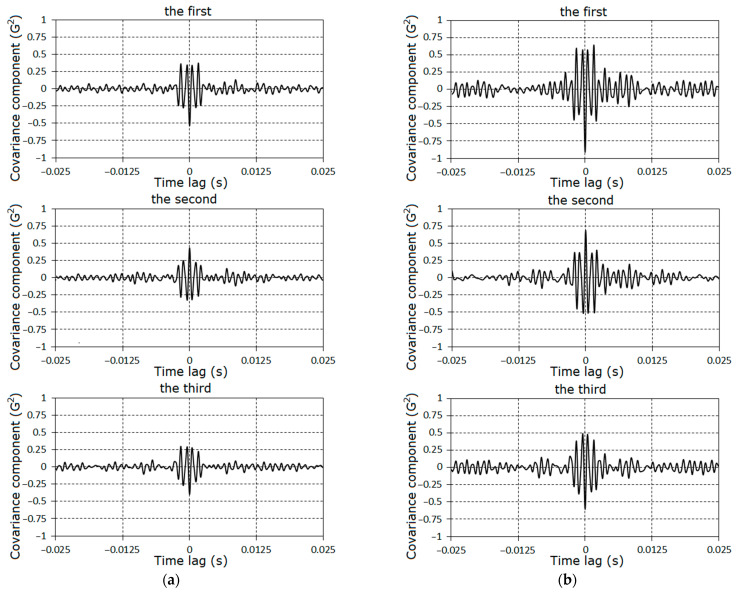
The dependences of the first, second, and third cosine covariance components estimators on starting lag for the second (**a**) and third (**b**) stages.

**Figure 20 sensors-21-06138-f020:**
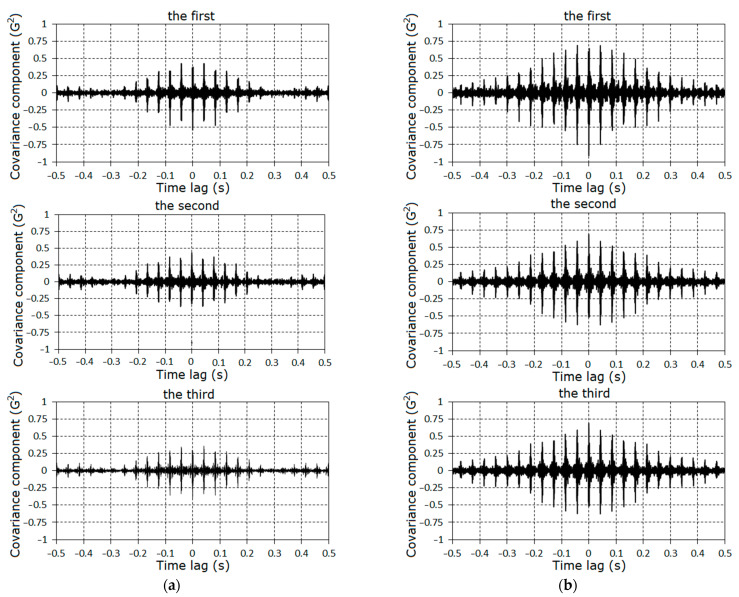
The dependences of the first, second, and third cosine covariance components estimators on the time lag for the second (**a**) and third (**b**) stages.

**Figure 21 sensors-21-06138-f021:**
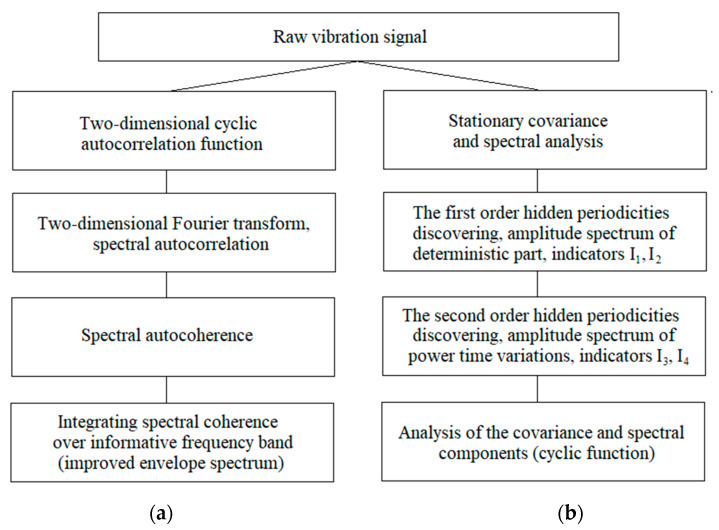
The main stages of the cyclostationary (**a**) and PCRP (**b**) analysis.

**Table 1 sensors-21-06138-t001:** Amplitudes of the deterministic oscillations harmonics.

Stage 1			Stage 2			Stage 3		
Orders	Frequency, Hz	$\hat{A}\left(k{\hat{f}}_{0}\right)$	Orders	Frequency, Hz	$\hat{A}\left(k{\hat{f}}_{0}\right)$	Orders	Frequency, Hz	$\hat{A}\left(k{\hat{f}}_{0}\right)$
0	0	0.000	0	0.00	0.000	0	0.00	0.000
1	24.20	0.045	1	24.05	0.039	1	23.41	0.080
2	48.40	0.020	2	48.10	0.011	2	46.82	0.004
3	72.60	0.002	3	72.15	0.003	3	70.23	0.008
4	96.80	0.007	4	96.20	0.029	4	93.64	0.047
5	121.00	0.041	5	120.25	0.198	5	117.05	0.132
6	145.20	0.079	6	144.30	0.330	6	140.46	0.447
7	169.40	0.240	7	168.35	1.062	7	163.87	1.278
8	193.60	0.409	8	192.40	1.654	8	187.28	1.677
9	217.80	0.115	9	216.45	0.546	9	210.69	0.679
10	242.00	0.038	10	240.50	0.162	10	234.10	0.049
11	266.20	0.065	11	264.55	0.265	11	257.51	0.335
12	290.40	0.005	12	288.60	0.141	12	280.92	0.275
13	314.60	0.116	13	312.65	0.270	13	304.33	0.193
14	338.80	0.231	14	336.70	0.382	14	327.74	0.518
15	363.00	0.191	15	360.75	0.405	15	351.15	0.286
16	387.20	0.165	16	384.80	0.345	16	374.56	0.486
17	411.40	0.143	17	408.85	0.260	17	397.97	0.319
18	435.60	0.064	18	432.90	0.163	18	421.38	0.333
19	459.80	0.030	19	456.95	0.068	19	444.79	0.122
20	484.00	0.010	20	481.00	0.018	20	468.20	0.070
21	508.20	0.010	21	505.05	0.019	21	491.61	0.060
22	532.40	0.029	22	529.10	0.029	22	515.02	0.048
23	556.60	0.030	23	553.15	0.045	23	538.43	0.050
24	580.80	0.014	24	577.20	0.020	24	561.84	0.036
25	605.00	0.519	25	601.25	0.230	25	585.25	0.339
26	629.20	0.005	26	625.30	0.023	26	608.66	0.062
27	653.40	0.014	27	649.35	0.010	27	632.07	0.040
28	677.60	0.006	28	673.40	0.021	28	655.48	0.027
29	701.80	0.018	29	697.45	0.012	29	678.89	0.038
30	726.00	0.038	30	721.50	0.023	30	702.30	0.062
31	750.20	0.062	31	745.55	0.039	31	725.71	0.171
32	774.40	0.063	32	769.60	0.032	32	749.12	0.156
33	798.60	0.040	33	793.65	0.020	33	772.53	0.095
34	822.80	0.053	34	817.70	0.018	34	795.94	0.077
35	847.00	0.089	35	841.75	0.006	35	819.35	0.147
36	871.20	0.048	36	865.80	0.016	36	842.76	0.240
37	895.40	0.062	37	889.85	0.034	37	866.17	0.183
38	919.60	0.055	38	913.90	0.039	38	889.58	0.182
39	943.80	0.047	39	937.95	0.039	39	912.99	0.170
40	968.00	0.080	40	962.00	0.084	40	936.40	0.182
41	992.20	0.061	41	986.05	0.078	41	959.81	0.246
42	1016.40	0.028	42	1010.10	0.061	42	983.22	0.216
43	1040.60	0.032	43	1034.15	0.050	43	1006.63	0.129
44	1064.80	0.010	44	1058.20	0.058	44	1030.04	0.110
45	1089.00	0.015	45	1082.25	0.028	45	1053.45	0.058
46	1113.20	0.021	46	1106.30	0.021	46	1076.86	0.047
47	1137.40	0.017	47	1130.35	0.020	47	1100.27	0.037
48	1161.60	0.010	48	1154.40	0.005	48	1123.68	0.019
49	1185.80	0.009	49	1178.45	0.022	49	1147.09	0.031
50	1210.00	0.005	50	1202.50	0.010	50	1170.50	0.010

**Table 2 sensors-21-06138-t002:** Amplitudes of the variances time variety.

Stage 1			Stage 2			Stage 3		
Orders	Frequency, Hz	$\hat{V}\left(k{\hat{f}}_{0}\right)$	Orders	Frequency, Hz	$\hat{V}\left(k{\hat{f}}_{0}\right)$	Orders	Frequency, Hz	$\hat{V}\left(k{\hat{f}}_{0}\right)$
0	0	0.223	0	0.00	0.721	0	0.00	0.991
1	24.20	0.036	1	24.05	0.541	1	23.41	0.949
2	48.40	0.024	2	48.10	0.435	2	46.82	0.702
3	72.60	0.027	3	72.15	0.419	3	70.23	0.617
4	96.80	0.016	4	96.20	0.402	4	93.64	0.605
5	121.00	0.011	5	120.25	0.389	5	117.05	0.600
6	145.20	0.015	6	144.30	0.364	6	140.46	0.544
7	169.40	0.016	7	168.35	0.290	7	163.87	0.421
8	193.60	0.023	8	192.40	0.236	8	187.28	0.288
9	217.80	0.017	9	216.45	0.162	9	210.69	0.301
10	242.00	0.012	10	240.50	0.113	10	234.10	0.224
11	266.20	0.007	11	264.55	0.096	11	257.51	0.180
12	290.40	0.004	12	288.60	0.104	12	280.92	0.149
13	314.60	0.008	13	312.65	0.062	13	304.33	0.089
14	338.80	0.013	14	336.70	0.034	14	327.74	0.108
15	363.00	0.002	15	360.75	0.051	15	351.15	0.083

**Table 3 sensors-21-06138-t003:** Degrees of fault development.

Degree	Initial	Small	Moderate	High	Emergency
$I_{2}$	<0.5	≥0.5 <2.0	≥2.0 <4.0	≥4.0 <10.0	≥10.0
$I_{4}$	<2.0	≥2.0 <10.0	≥10.0 <20.0	≥20.0 <25.0	≥25.0

## Data Availability

The data used in these study are available on request from the corresponding author.

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
