# Peer review of "Methods of Hidden Periodicity Discovering for Gearbox Fault Detection"

_sensors, 2021, doi:10.3390/s21186138_

Round 1
Reviewer 1 Report
This paper addresses an interesting problem on gearbox fault detection, the authors proposed a model for analysis of damaged gear pair. Overall, the paper is well organized. However, several issues still need to be improved before further consideration.
- The problem solved in this paper and advantages of the results should be described briefly in the abstract. In order to ensure that the readers can understand the research content of the article, the author is advised to revise the abstract.
- Please add legends to Figure 13 and 17 so as to illustrate the corresponding stages of the lines. Please adjust the resolution of Figure 14 (a) and (b) and make them consistent.
- Please add content to fully illustrate the significance of figures 19 and 20.
- On line 419 of page 13, a parenthesis is missing.
- Please try to add the comparison with other diagnosis methods to highlight the advantages of proposed method in section 4. And the results should be as quantitatively as possible.
- The English of the manuscript must be improved before resubmission. We strongly suggest that you obtain assistance from a colleague who is well-versed in English or whose native language is English.
Author Response
We thanks to reviewers for their detailed analysis of the manuscript and valuable remarks. We took into account all comments and made respective corrections.
Our answers to reviewer’s comments are given below:
Reviewer 1
- “The problem solved in this paper …. to revise the abstract”
We present revised abstract.
- “Please add legends … them consistent”
We add legends to Figures 13 and 17 and introduce amendments to Figures 14a and 14b.
- “Please add content ….figures 19 and 20”
We add content for Figures 19 and 20.
- “On line 419 … is missing”
Parenthesis is corrected.
- “Please try to add …as possible”
We add the comparison of the different indicators which can be formed on the basis of the obtained numerical results in Discussion.
- “The English ….whose native language is English”.
English was revised by Proof-Reading Service (http://www. proof-reading service.com)
Reviewer 2 Report
The paper proposes a method of hidden periodicity discovering for gearbox fault detection. A model in the form of the BPCRP is proposed to analyze the vibrations of a damaged gear pair. However, there are follows questions that to be considered.
Minor Revision
1. The formulas in lines 168 and 172 are unnumbered.
- The words (e.g. modelling/modeling, Figure/fig) needs to be unified.
- The names of each diagram in all Figs need to be explained and the units of axes for each diagram need to be clearly explained.
Major Revision
- In the abstract, the background and significance of the research are not introduced.
- In Sections 2 and 3, the author should majorly introduce the work, which is creative.
- How to classify and determine the different stages of gear tooth failure in Figs 4 and 5.
- The author needs to clarify how the coefficients of the formula in the line 419 are obtained.
- The manuscript should be refined to highlight the originality and significance of the work done by the author.
Author Response
We thanks to reviewers for their detailed analysis of the manuscript and valuable remarks. We took into account all comments and made respective corrections.
Our answers to reviewer’s comments are given below:
Reviewer 2
Minor Revision
- “The formulas …..are unnumbered”
There is not reference below to formula in line 168.
The formula in line 172 is numbered now (4).
- “The words ….to be unified”
The words are unified.
- “The name of each diagram ….clearly explained”
All figures and diagrams are explained now.
Major Revision.
- “In the abstract ….are not introduced”
Abstract is revised.
- “In Section 2 and 3 ….is creative”
In Sections 2 and 3 ours new theoretical results are presented. These results were not published ealier. In ours point of view they are significant assistance for readers to understand and use in the future the developed techniques. Such applications are not known in literature.
- “How to classify ….in Figs. 4 and 5”
We describe the specific features of Figure 4 and Figure 5 for different stages.
- “The author needs …are obtained”
Hamming window is widely used for spectral analysis of the time series. The numerical values of the window parameters are chosen to obtain optimal relationship between the parameters of the main and side maximums of the spectral window (see, for example, J. Max. Methodes et techniques de traitement du signal et applications aux measures physiques. Paris, 1981.)
5. “The manuscript should be … by the author”
It was noted above, that in the paper only our unpublished results are presented. Combined account of the theoretical basing of the PCRP methods for the covariance analysis of time series with unknown non-stationary period and the results of their applications for fault detection was the main purpose of the paper.
Reviewer 3 Report
The paper is very extensive and includes many developments so that sometims it is even too detailed in analyzing methods an links amon them. But it is a minor limit. Enclosed a file with some remarks about miprints or other minor observations.

Author Response
We thanks to reviewers for their detailed analysis of the manuscript and valuable remarks. We took into account all comments and made respective corrections.
Our answers to reviewer’s comments are given below:
Reviewer 3.
Abstract
The new version of the abstract is presented.
- Introduction
The following corrections are made:
Line 62; Line 77; Line 78; Line 95; Line 125; Line 127; Line 142; Line 148.
They are marked.
The description of the paper content is reduced.
- BPCRP as a model of gear pair vibration signal.
Following corrections are made:
Line 164; Line 77; Line 172; Line 189; Line 191; Line 214; Line 222; Line 226; Line 230; Line 240.
3.Gear fault detection as PCRP estimation issue
Following corrections are made:
Line 289; Line 293; Line 295; Line 331; Lines 341-342; Line 345; Line 349; Line 356; Line 387; Line 390; Line 393; Line 398.
- The analysis of the natural data
Following corrections are made:
Line 405; Line 459; Line 479; Line 486; Lines 575-576; Line 578; Line 581.
- Discussion
Following corrections are made:
Line 664; Line 674; Line 705; Line 719; Line 740.
- Conclusions
Following corrections are made:
Line 756; Lines 764-768; Line 796; Lines 809-810.
The first part of the conclusion is reduced.
Author contribution
Following correction are made: Line 821
References
- Line 837: Ukrain → Ukrainian
- Ref. # 40 is the monograph section.
- In Refs. # 55 and #57 pages are not yet available.
We ask You to consider the corrected version of our paper.
Round 2
Reviewer 1 Report
The authors have made corresponding modifications and supplements. I have no more questions.
Reviewer 2 Report
The response has answered the questions and the manuscript has been revised as well. The manuscript could be published in the journal after revising some formula editing errors.